# Statistical study of linear magnetic hole structures near Earth

Martin Volwerk[1], David Mautner[1], Cyril Simon Wedlund[1], Charlotte Goetz[2], Ferdinand Plaschke[1], Tomas Karlsson[3], Daniel Schmid[1], Diana Rojas-Castillo[1,4], Owen W. Roberts[1], and Ali Varsani[1]

[1]Space Research Institute, Austrian Academy of Sciences, Graz, Austria
[2]ESTEC, European Space Agency, Keplerlaan 1, 2201AZ Noordwijk, The Netherlands
[3]Department of Space and Plasma Physics, School of Electrical Engineering and Computer Science, Royal Institute of Technology, Stockholm, Sweden
[4]Instituto de Geofísica, UNAM, Ciudad Universitaria, Coyoacán, CP 04510, Ciudad de México, México

**Correspondence:** M. Volwerk, Space Research Institute, Austrian Academy of Sciences, Schmiedlstr. 6, 8042 Graz, Austria (martin.volwerk@oeaw.ac.at)

**Abstract.** The MMS1 data for 8 months in the winter periods of 2017-2018 and 2018-2019, when MMS had its apogee in the upstream solar wind of the Earth's bow shock, are used to study Linear Magnetic Holes (LMHs). These LMHs are characterized by a magnetic depression of more than 50% and a rotation of the background magnetic field of less then $10°$. 406 LMHs are found and, based on their magnetoplasma characteristics, are split into three categories: cold (increase in density, little change in ion temperature), hot (increase in ion temperature, decrease in density) and sign change (at least one magnetic field component changes sign). The occurrence rate of LMHs is 2.3 per day. All LMHs are basically in pressure balance with the ambient plasma. Most of the linear magnetic holes are found in ambient plasmas that are stable against the mirror-mode generation, but only half of the holes are mirror-mode stable inside.

## 1 Introduction

One of the structures in the Interplanetary Magnetic Field (IMF) that can be found throughout the solar system, is the Magnetic Hole (MH), first discussed by Turner et al. (1977). These are depressions in the magnetic field strength up to 90% of the background magnetic field. Although Turner et al. (1977) did not have adequate plasma measurement, they assumed that these were diamagnetic structures. Theoretically this was discussed by Burlaga and Lemaire (1978) and later measurements have shown that increased plasma pressure in the MHs takes care of the pressure balance (see e.g., Burlaga et al., 1990; Winterhalter et al., 1995). A special case of MHs where the magnetic field direction does not change more than $10°$ is called Linear Magnetic Hole (LMH, Turner et al., 1977)).

The origin of (L)MHs is still not completely clear. These structures appear in the solar wind for high plasma-$\beta$ conditions, and they are most likely related to mirror modes (MMs) or might even be the end stage of MMs (Winterhalter et al., 1994).

Winterhalter et al. (2000), for example, found that trains of MMs were observed in MM unstable ambient plasma, but when the instability criterion was not fulfilled for the ambient plasma there were only LMHs.

MMs occur in high-$\beta$ plasmas with a temperature asymmetry $T_\perp > T_\parallel$ (Gary et al., 1993) and specifically the instability criterion for a bi-Maxwellian distribution is (Southwood and Kivelson, 1993):

$$R_{\mathrm{SK}} = \frac{T_{i\perp}/T_{i\parallel}}{1 + 1/\beta_{i\perp}} > 1, \tag{1}$$

where

$$\beta_{i\perp} = \frac{n_i k_{\mathrm{B}} T_{i\perp}}{B^2/2\mu_0}, \tag{2}$$

with $T_{i\perp}$ and $T_{i\parallel}$ the perpendicular and parallel ion temperature, $n_i$ the ion density, $B$ the magnetic field strength, $k_{\mathrm{B}}$ Bolzmann's constant and $\mu_0$ the permeability of vacuum. (for a more general discussion of the instability criterion see e.g. Hellinger, 2007). Stevens and Kasper (2007) found that LMHs mainly occurred in MM-stable regions ($R_{\mathrm{SK}} < 1$). This might lead to the conclusion that as soon as the plasma becomes MM-stable the MMs start to diffuse/transform into MHs. Hasegawa and Tsurutani (2011) proposed a Bohm-like diffusion (Bohm et al., 1949) process taking place in the MMs, where the higher frequencies of the structure decay faster than the lower frequencies, and thereby the MMs grow in size as they move away from the generation location. The scale size of the MMs is then given by:

$$\lambda(\mathcal{L}) = \lambda_0 \left( 1 + \frac{\omega_{ic}\mathcal{L}}{32u} \right), \tag{3}$$

where $\lambda_0$ is the MM size at the source region, $\mathcal{L}$ is the distance from the source region, $\omega_{ic}$ is the ion cyclotron frequency and $u$ is the convection speed of the MMs. Schmid et al. (2014) showed that the growth of MMs in Venus's and comet 1P/Halley's magnetosheaths was well described by this process for pick-up ions species protons and water, respectively. It can be envisioned that through the growth of the MMs the wave trains merge into larger structures leading to MHs.

There have been many studies on the occurrence rate of LMHs throughout the solar system. In Table 1 the rates are listed from the inner solar system to the outer reaches. It is clear that most occurrence rates are about a few per day. However, it is difficult to compare the exact values listed. As an example, it is immediately obvious that occurrence rates differ strongly near Venus: Zhang et al. (2008) found a rate of 4.5 per day, whereas Volwerk et al. (2020) reported a rate of 1.0 per day. These studies were done for 2006 and 2007, respectively; the difference was not caused by changes in solar activity but by the selection criteria used: $\Delta B/B > 0.25$ vs $0.50$ $\Delta\theta < 15°$ vs $10°$. This shows the necessity of studying these structures with one unified set of conditions, only then can precise physical statements be made about their characteristics and development (see also Klein and Burlaga, 1980).

In this paper the Magnetospheric MultiScale (MMS, Burch et al., 2016) mission is used, for the winter seasons of 2017/18 and 2018/19, when the spacecraft had their apogee in the upstream solar wind. A statistical study is done for the occurrence rate and the characteristics of the LHM structures that are found in the data.

**Table 1.** Occurrence rates of magnetic holes in the solar wind throughout the system.

| Location | Occurrence Rate (per day) | Spacecraft | # events / # days / years | Reference |
|---|---|---|---|---|
| Mercury | 4.4 | Messenger | 2726 / 618 / 2011 - 2015 | Karlsson et al. (2020) |
| Mercury - Venus | 3.3 - 1.0 | Messenger | 96 / 1140 / 2007 - 2011 | Volwerk et al. (2020) |
| Venus | 4.2 | VEX | 791 / 189 / 2006 | Zhang et al. (2008) |
| Mercury - Earth | 2.2 - 1.7 | Helios 1 & 2 | 601 / - / 1974 - 1976 | Sperveslage et al. (2000) |
| Earth L1 | 0.6 | WIND | 2074 / - / 1994 - 2004 | Stevens and Kasper (2007) |
| Earth | 1.5 | IMP I [1] | 28 / 18 / 1971 | Turner et al. (1977) |
| Earth | 1.8 | Cluster | 897 / - / 2001 - 2009 | Xiao et al. (2014) |
| Earth | 2.3 (0.9) [2] | MMS | 406 / 177 / 2017 - 2019 | *this study* |
| Mars | 2.1 | MAVEN | 102 / 56 / 2016 | Madanian et al. (2020) |
| 2 - 11 AU | $\sim 0.5 \rightarrow 0.1$ | Voyager 2 | 235 / - / 1978 - 1982 | Sperveslage et al. (2000) |
| 11 - 17 AU | $\sim 0.1$ | Voyager 2 | 16 / - / 1982 - 1985 | Sperveslage et al. (2000) |
| High Solar Latitude | 5.2 | Ulysses | 4127 / 780 / 1990 - 1992 | Winterhalter et al. (2000) |
| Heliosheath | 2 - 3 | Voyager 1 | 24 / 9 / 2006 | Burlaga et al. (2007) |

## 2 Instrumentation and Data Analysis

The MMS FGM fast data (Russell et al., 2016) are used with a sampling rate of 16 Hz, which are down-sampled to 1-s resolution. The data are restricted to such time intervals that MMS had its apogee in the solar wind, i.e. November 2017 through March 2018 and December 2018 through March 2019. An additional limit was set to the location of the spacecraft, namely that it be farther out than 15 $R_{\mathrm{E}}$, to avoid influence from the Earth's bow shock, which can move further outward in times of low solar wind pressure (Meziane et al., 2014). However this will not exclude the foreshock region from the data set, which extends much further upstream (Heppner et al., 1068; Greenstadt et al., 1968; Scarf et al., 1970).

The FGM data was then handled as in previous papers by Plaschke et al. (2018) and Volwerk et al. (2020) in order to find linear magnetic holes (LMHs);

1. The background magnetic field, $B_{300}$, is determined by a sliding window average over 300 s;

2. The data are smoothed by a sliding window average of 2 s, which gives $B_2$;

3. The ratio $\Delta B/B_{300} = (B_{300} - B_2)/B_{300}$ is calculated and the lowest field depressions are selected that are at least 300 s apart;

4. The background field should be $B_{300} \geq 2$ nT;

5. The ratio $\Delta B/B_{300} > 0.5$;

6. The rotation of the magnetic field over the hole $\theta \leq 10°$.

All data are in the Geocentric Solar Equatorial (GSE) coordinates. This resulted in 406 LMH structures observed between 15 and 30 $R_{\mathrm{E}}$ from the Earth, an example of which can be seen in Fig. 1, and for all structures plasma data is available.

As compared with previous papers (Plaschke et al., 2018; Volwerk et al., 2020), MMS has plasma data from the Fast Plasma Instrument (FPI Pollock et al., 2016), with a temporal resolution for the fast mode of 4.5 s for ions and electrons. These measurements allow for determining the ion density and temperature inside the LMH structures. However, the FPI moments need to be looked at carefully, because the FPI instrument was not developed for solar wind conditions. This means that the different parameters that are obtained from the instrument may be inaccurate. In order to check the quality of the FPI data, they can be compared with data from solar wind-specialized missions such as Wind (Lin et al., 1995). Something similar occurs with the ARTEMIS mission (Angelopoulos, 2011), which has basically the same plasma instrument and only has a magnetospheric mode. Artemyev et al. (2018) compared 6 years of ARTEMIS data with the OMNI data set (King and Papitashvili, 2005). They found good correlation, within a factor 2, between the ARTEMIS electron density and the OMNI ion density.

Bandyopadhyay et al. (2020) compared the FPI data with Wind for one event. They found that there was a slight discrepancy for the proton density, whereas the proton velocity was in good agreement and the proton temperature was underestimated. Recently, Roberts et al. (2021, in preparation) presented a statistical study comparing the ion and electron data from FPI in the solar wind with the OMNI data. They found that the ion density was underestimated (up to a factor 2 for densities greater than 10 $\mathrm{cm}^{-3}$) and the ion temperature was overestimated (up to a factor 2). However, for the electrons they found good agreement between the two data sets assuming quasi-neutrality of the plasma to calculate the OMNI electron density from the OMNI ion density.

As FPI was not specifically developed for solar wind conditions, this also means that there can be spurious signals in the FPI data, such that the spin tone at $\sim 20$ s is not removed correctly. This will then appear as $\sim 20$ s variation in various plasma components such as density and velocity.

In this paper only the MMS1 data will be used, as the inter-spacecraft distance is too small to show any significant differences between the four spacecraft with respect to the structures that are investigated. Also, as there is only burst mode data (at a resolution of 30 ms) for a small number of the identified structures the fast mode FPI data are used in this paper.

Taking together all the LMH structures and determining the dwelling time of MMS in the region between 15 and 30 $R_{\mathrm{E}}$ gives an occurrence rate of 2.3 per day. This is close to what was found in previous studies: 1.5 per day Turner et al. (1977), 1.7 - 2.2 per day Sperveslage et al. (2000), 0.6 per day Stevens and Kasper (2007) and 1.8 per day Xiao et al. (2014) (see also Table 1).

The occurrence rate is calculated as a function of the distance from Earth and shown in Fig. 2 with blue bars (the other colours are defined further below). There is a strong variation in the occurrence rates, showing the randomness with which these structures appear.

The apparent temporal width, $w$, of the LMH structures in the time series is also determined during the search for the events, which corresponds to the Full Width at Half Maximum (FWHM). In Fig. 3 the distribution is shown, which peaks in the bins

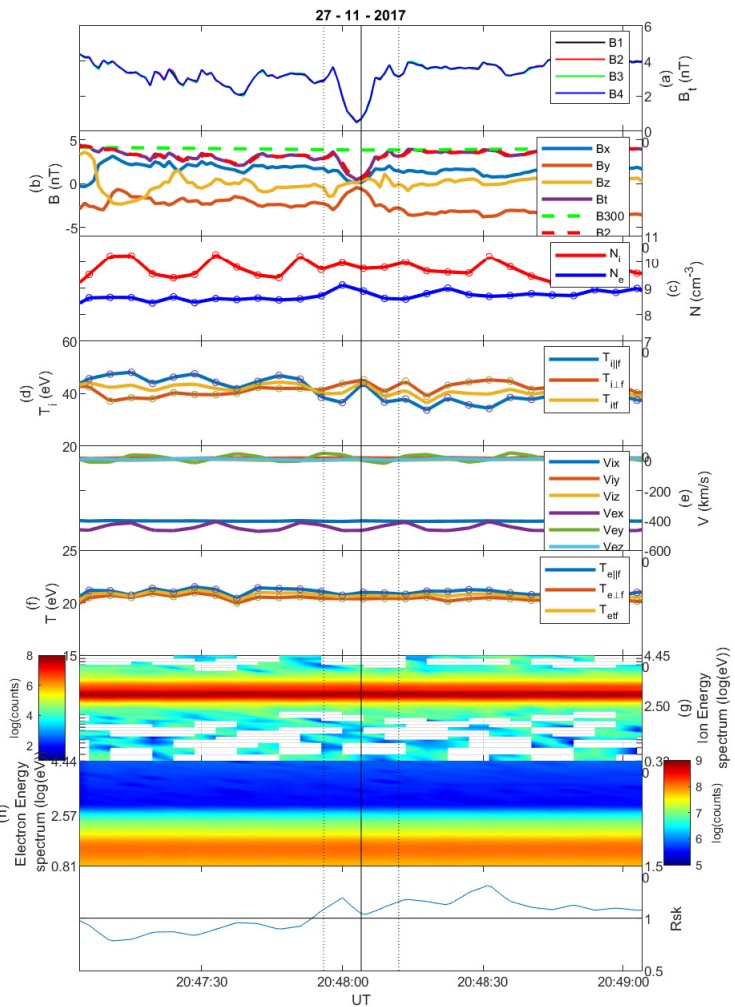

**Figure 1.** Linear Magnetic Hole on 27 November 2017, category "cold". (a) The total magnetic field $B_\mathrm{t}$ for all four MMS spacecraft; (b) The magnetic field components with $B_{300}$ and $B_2$; (c) The fast mode ion and electron density; (d) The parallel, perpendicular and total ion temperature in fast mode; (e) The fast mode ion and electron velocity components. (f) The parallel, perpendicular and total electron temperature in fast mode; (g) The ion energy spectrogram; (h) The electron energy spectrogram; (i) The instability criterion $R_\mathrm{SK}$.

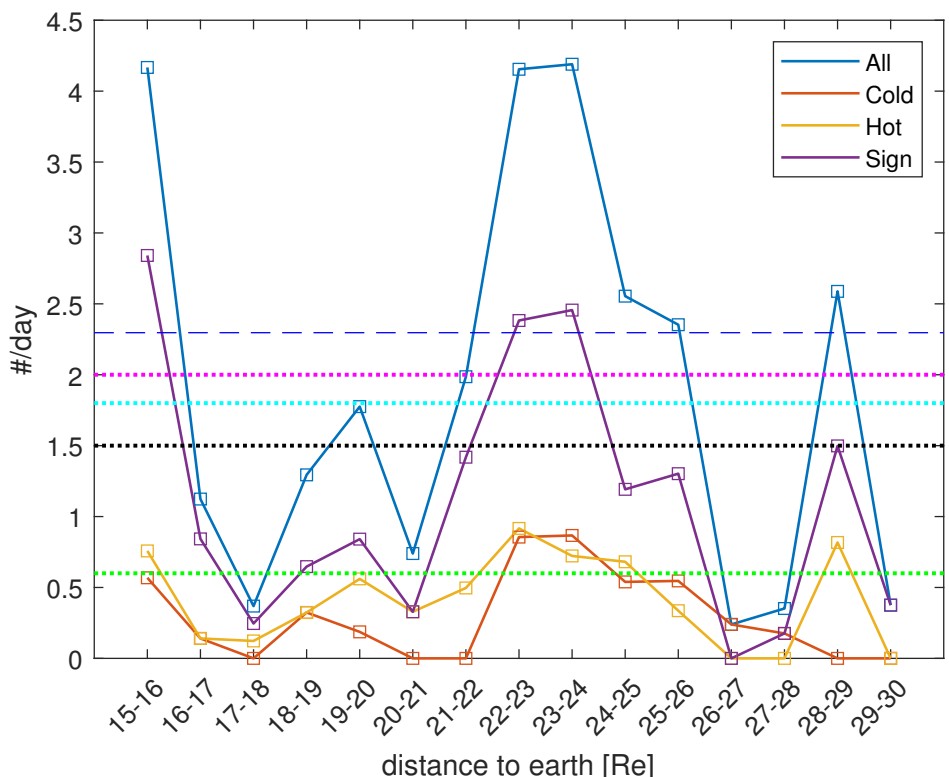

**Figure 2.** The occurrence rate of LMH structures as a function of the distance from Earth. The blue squares connected by blue lines show the occurrence rates for all structures. The red squares and line for the cold LMHs; the yellow for the hot LMHs; and the purple for sign-change LMHs. The dashed blue line is the average occurence rate (2.3 per day) and the coloured dotted lines are the daily occurrence rates from literature: 1.5 (black Turner et al., 1977), 2.0 (magenta Sperveslage et al., 2000), 0.6 (green Stevens and Kasper, 2007) and 1.8 (cyan Xiao et al., 2014).

5-10 and 10-15 seconds. This value agrees well with the highest occurrence of widths found between Mercury and Venus (Sperveslage et al., 2000; Volwerk et al., 2020).

Of course, the width, $w$, of the structures measured in seconds does not say anything about the physical size of the structures, and is mostly used when plasma parameters are not available for analysis. In the case of MMS the plasma data can be used to transform the width, $w$, into a physical size $L$. In Fig. 4 the size is given in units of the local proton thermal Larmor radii $\rho_\mathrm{L}$:

$$L = \frac{w * V_\mathrm{SW}}{\rho_\mathrm{L}} = \frac{w V_\mathrm{SW} q_p B}{\sqrt{2 q_p m_p T_{p\perp}}}, \tag{4}$$

where the perpendicular thermal velocity of the ions is is calculated from the ion temperature ($v_{\mathrm{th}\perp} = \sqrt{k_B T_{p\perp}/m_p}$). The distribution for all events peaks in the bins $L = 5 - 20\rho_\mathrm{L}$.

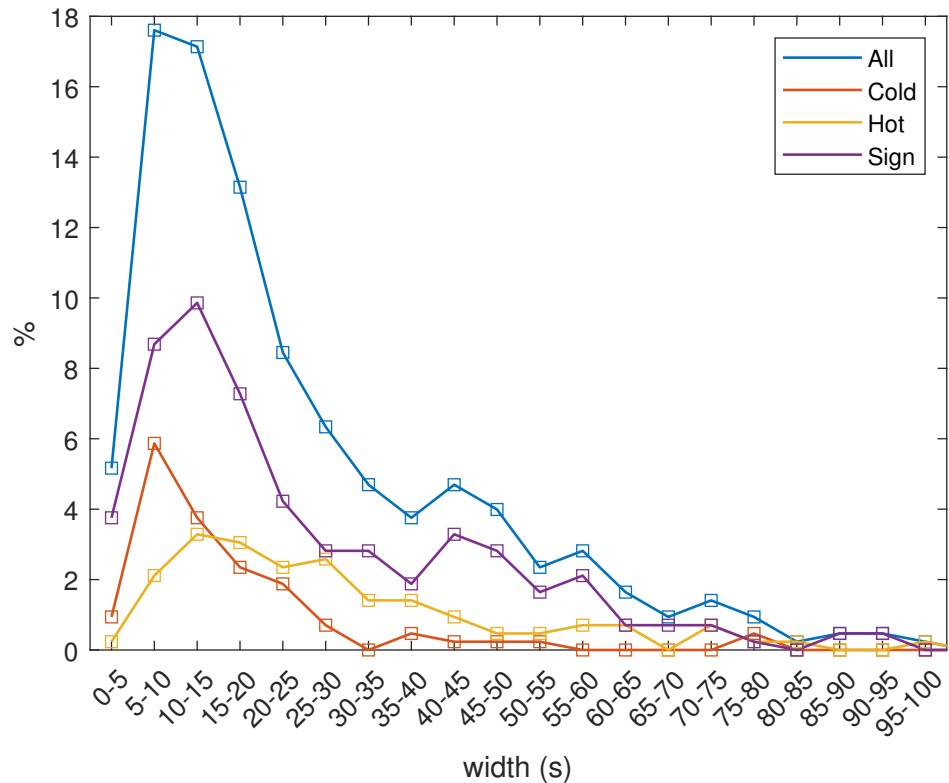

**Figure 3.** The distribution of the widths of the LMH structures in % of total events. The colour coding is the same as in Fig. 2

## 3 Categorization

The identification of the LHM structures was done in the same way as in previous papers. The vicinity of the Earth and its bow shock could have an influence on the structures that are selected. Possibly, foreshock structures whose magnetic field signatures resembles a magnetic depletion like hot flow anomalies (e.g., Schwartz, 1995; Zhao et al., 2017), cavitons (Kajdič et al., 2013), cavities (Sibeck et al., 2002, 2004), density holes (Parks et al., 2006) or foreshock bubbles (Turner et al., 2020) can disturb the determination occurrence rate of MHs. Therefore, a categorization of these structures is made based on their magnetic and plasma characteristics, which will lead to 3 kinds of structures, discussed in the following subsections.

Fig. 5 shows a scatter plot of all structures, described by $\Delta N/N = (N_{\text{in}} - N_{\text{out}})/N_{\text{out}}$ and $\Delta T/T = (T_{\text{in}} - T_{\text{out}})/T_{\text{out}}$ and colour labeled with $\Delta B/B$. The figure shows that only a minority of the structures has $\Delta N/N > 0$, whereas the majority of events has $\Delta T/T > 0$. This means that pressure balance can be obtained in different ways: an increase in density or in temperature (or both). Noticeable is the almost empty upper right quadrant in the panels.

Three categories are defined below which are visualized in Fig. 5 (b-d): cold ($\Delta N/N > 0$), hot ($\Delta N/N \leq 0, \Delta T/T > 0$), sign change (remaining cases, where one B component changes signs).

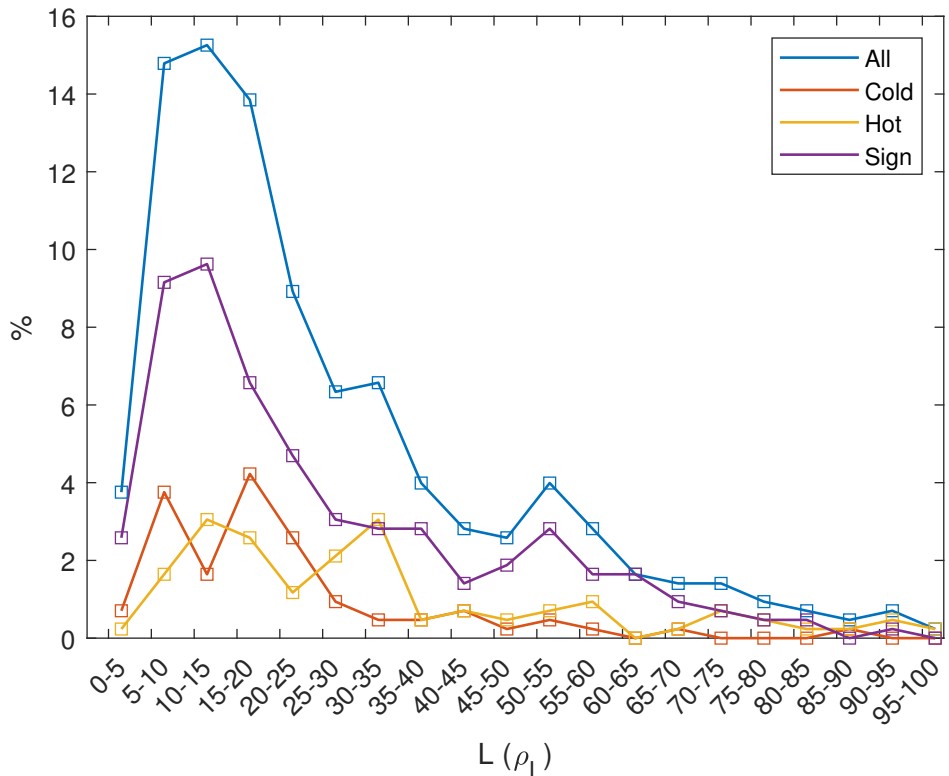

**Figure 4.** The distribution of the size $L$ of the LMH structures in % of total events. The colour coding is the same as in Fig. 2

## 3.1 Cold LMH

The cold LMH is defined by a decrease in magnetic field strength and an increase in density, which leads to pressure balance over the structure, as shown in Fig. 5(b). The structure in Fig. 1 is an example, and overall there are 75 structures that fall into this category. In Figs. 2, 3 and 4 these structures are represented by orange lines, which do not seem to have a distribution significantly different from the whole set of structures (blue lines).

Fig. 1 shows an LMH with $\Delta B/B \approx 0.84$. The electron density increases, albeit slightly shifted with respect to the centre of the hole, and the ion density remains almost constant in the hole. The oscillations that are seen in the ion density (as well as in the temperature and the velocity), at $\sim 20$ s are caused by the spin tone of the spacecraft. The ion temperature shows that the parallel component $T_{i\parallel}$ increases in the middle of the hole, whereas the total ion temperature $T_{iT} = (T_{i\perp} + T_{i\parallel})/2$ only changes little. Before the hole in the solar wind $T_{i\perp} > T_{i\parallel}$, whereas inside the hole and after the hole in the solar wind $T_{i\perp} > T_{i\parallel}$. The electrons do not show any significant changes in temperature. The instabilitiy criterion $R_{SK}$ shows that before the hole the solar wind plasma was MM-stable, whereas inside and after the hole the plasma is MM-unstable.

Fig. 6 shows a very classical example of a LMH in the solar wind. There are actually 2 holes in this example, with the marked one being the deepest. Because of the selection criteria, the first one is not counted as an event, which will have influence on the

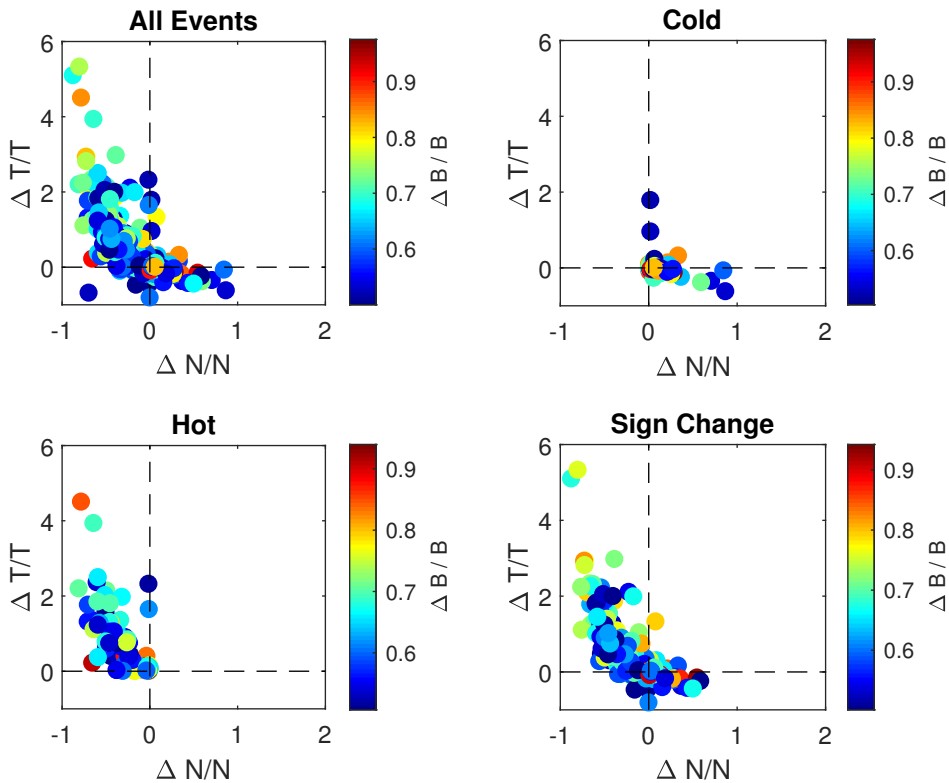

**Figure 5.** Scatter plot of the $\Delta T/T$ vs. $\Delta N/N$ for all LMH structures. The colour coding is the $\Delta B/B$ of each structure.

occurrence rate of MHs, resulting in a 10% difference in the total number of MHs (see Volwerk et al., 2020). The ion density
data show an increase in density inside the holes of around 1 to 2 cm$^{-3}$, whereas the electron density shows only little increase.
The total ion temperature remains rather constant over the whole time interval in the figure, with $T_{i\parallel} > T_{i\perp}$, and $R_{\mathrm{SK}} < 1$
i.e. MM-stable. The total electron temperature $T_{et} = (T_{e\perp} + T_{i\parallel})/2$ slightly decreases in the central hole, and in both holes
$T_{e\parallel} \approx T_{e\perp}$, but $T_{e\parallel}$ decreases in both holes.

### 3.2   Hot LMH

The hot LMH is defined by a decrease in density inside the hole (or a very small increase), and a strong increase in temperature,
Fig. 5(c). An example of a hot LMH is shown in Fig. 7. Once again there is a decrease of the magnetic field strength with
$\Delta B/B \approx 0.61$ with very little density variation, but a strong increase in $T_{i\perp}$. The hole is embedded in a MM-stable plasma,
with $R_{\mathrm{SK}} < 1$; only in the left part of the event window does it show a value $> 1$, where $T_{i\perp} > T_{i\parallel}$. Also interesting is the
"double dipped" magnetic structure, which could be an indication of the merging of two holes, which will be addressed in the
discussion section.

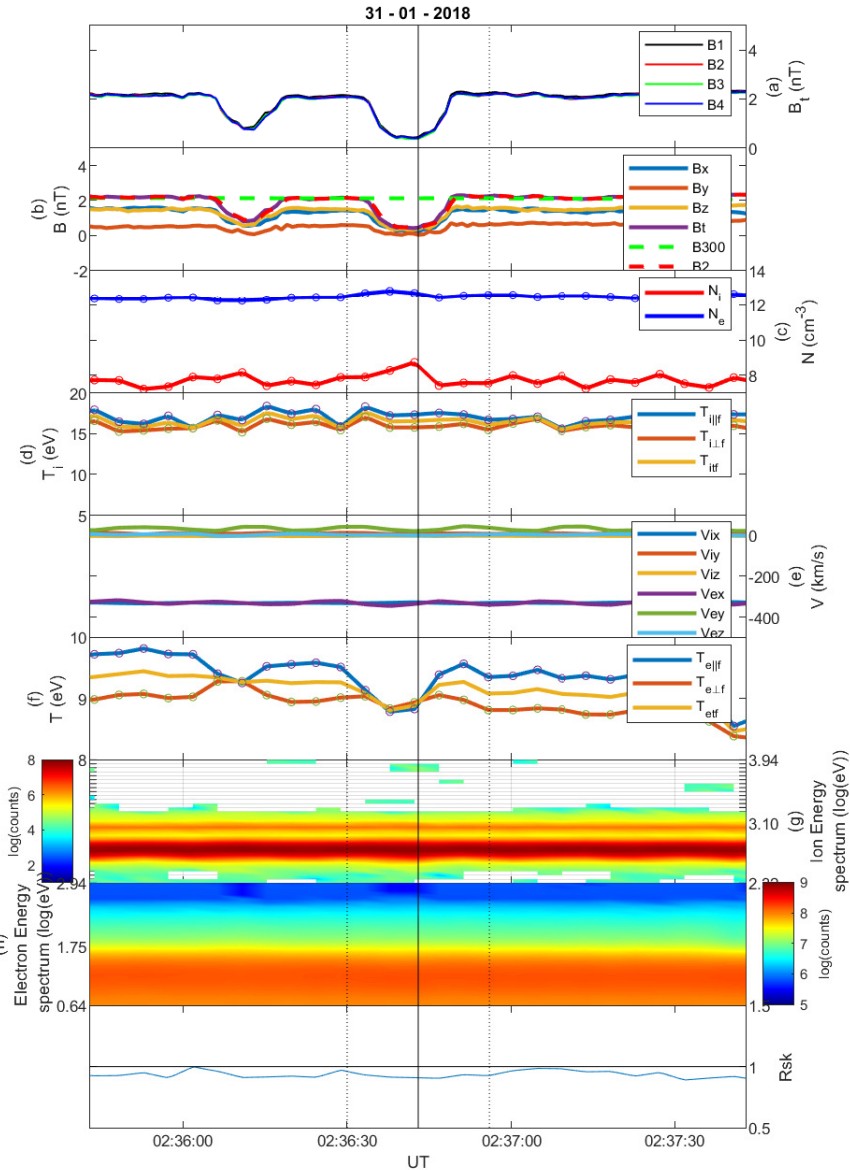

**Figure 6.** Linear Magnetic Hole on 31 January 2018, category "cold". Same format as in Fig. 1

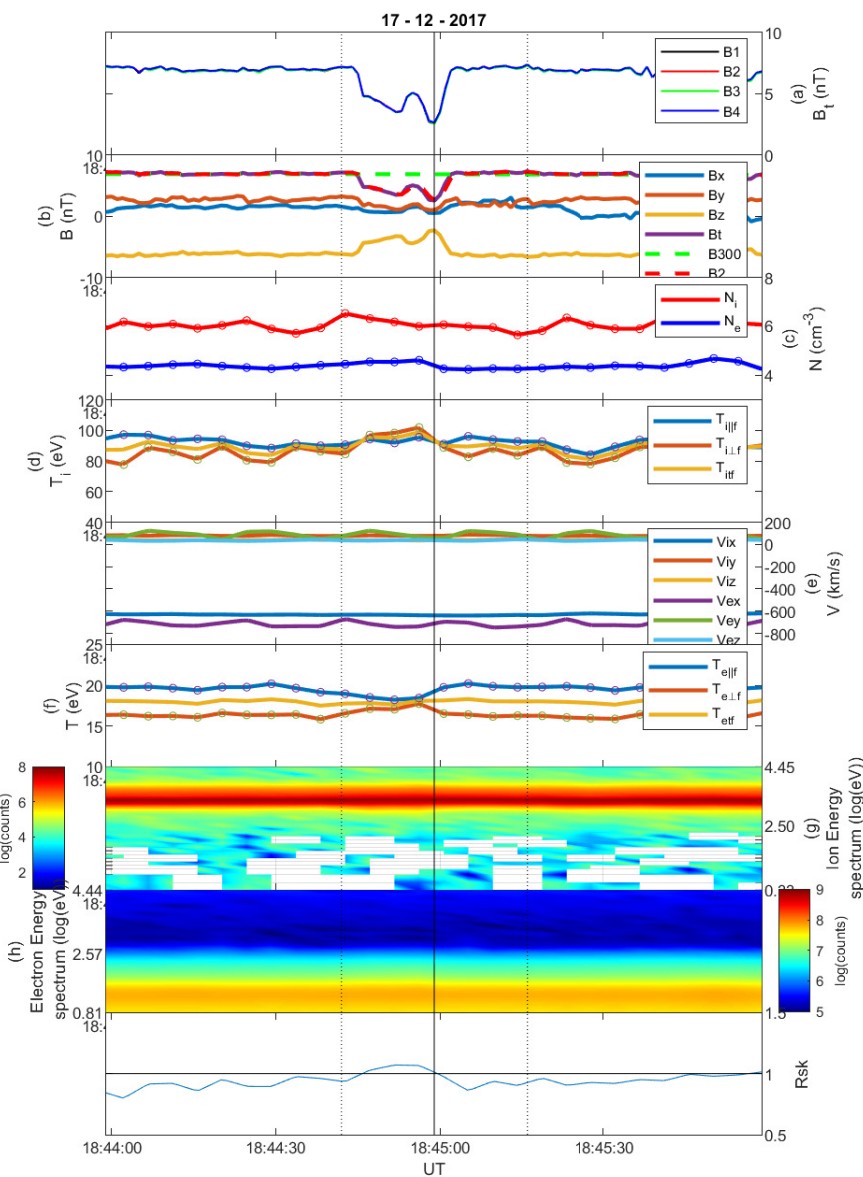

**Figure 7.** Linear magnetic hole on 17 December 2017, category "hot". Format as in Fig. 1.

In total there are 94 structures in this category, which are shown by yellow lines in Figs. 2, 3, and 4. They approximately follow the same distribution as the full set in Fig. 2. However, looking at their sizes in Figs. 3 and 4 the distribution looks slightly broader than for the other categories, up to $L = 35\rho_{\mathrm{L}}$.

In Fig. 8 another example of a hot hole is shown. The decrease in $B$ is combined with a constant density and an increase in temperature $T_{it}$, mainly created by an increase in $T_{i\perp}$. Overall, the perpendicular ion temperature remains greater than the parallel ion temperature and $R_{\mathrm{SK}} > 1$, so the region and the structures are MM-unstable until after the second dotted line. Only $\sim 30$ s after the LMH the plasma becomes MM-stable after an abrupt change in the IMF direction, a rotation from $B_y$ into $B_z$. The electrons show a decrease in $T_{e\parallel}$ inside the hole.

## 3.3 Sign Change LMH

After the definition of the "cold" and "hot" categories of LMHs, there is still a large group of events that is left. These all have the characteristic that (at least) one of the magnetic field components change sign over the structure. There are a total of 237 structures in this category, i.e. the majority. They show up both in the cold and hot quadrants in Fig. 5(d), 43 are "cold", and 155 are "hot", and 39 in the region where both the density and temperature decreases.

An example is shown in Fig. 9, and, although there is a sign change, the rotation of the field over the hole is still less than $10°$. There is a an increase in the ion temperature over the structure but the parallel electron temperature decreases, the density remains constant. Just outside the hole the ion temperature drops and the electron temperature and density increase.

There is possibly an additional MH at the beginning of the interval shown in Fig. 9, where at $\sim 0410 : 10$ UT, there is an increase in ion and electron density, $T_{i\perp}$ drops to be equal with $T_{i\parallel}$, but $T_{e\perp}$ increases slightly.

The solar wind is strongly deflected by $\sim 45°$ in the $XY_{\mathrm{GSE}}$ direction. The ion energy spectrum is very broad indicating that this event is most likely in the Earth's magnetosheath.

Fig. 10 shows another example of a sign-change LMH, where clearly $B_y$ and $B_z$ change signs over the width of the hole. The ion and electron density drop drastically, and the ion and electron temperature increase drastically. The solar wind is not deflected, as in the previous case, but the ion energy spectrum shows very hot ions above the narrow solar wind ions, indicating that this structure is also in the Earth's foreshock.

For the structure shown in Fig. 10 a minimum variance analysis (MVA, Sonnerup and Scheible, 1998) was performed on the interval between the two vertical dotted lines in panel (a), and the magnetic field data were transformed into the $lmn$ system (with $l$ for maximum, $m$ for intermediate and $n$ for minimum variance directions), shown in Fig. 11. The ratio of the intermediate-to-minimum eigenvalue is $\lambda_{\mathrm{int}}/\lambda_{\mathrm{min}} \approx 5.9$ whereas that of the maximum-to-intermediate eigenvalue is $\lambda_{\mathrm{max}}/\lambda_{\mathrm{int}} \approx 2.3$, which means that the MVA is well determined (Sergeev et al., 2006). Between the two maxima in $B_{\mathrm{t}}$ (purple) the $B_n$ component (blue) remains almost constant, $B_m$ (yellow) decreases strongly towards values around 0 nT, and $B_l$ shows a sign change from $\sim 5$ nT to $\sim -4$ nT. The $l$ direction is $[-0.52, 0.0, 0.8]$, mainly in the $Z_{\mathrm{gse}}$ direction, and the location of the spacecraft is $[20.6, 2.3, 6.5]R_E$. This behaviour may be consistent with the signature of a flux rope passing over the spacecraft, or of a hot flow anomaly (see e.g., Schwartz et al., 2018).

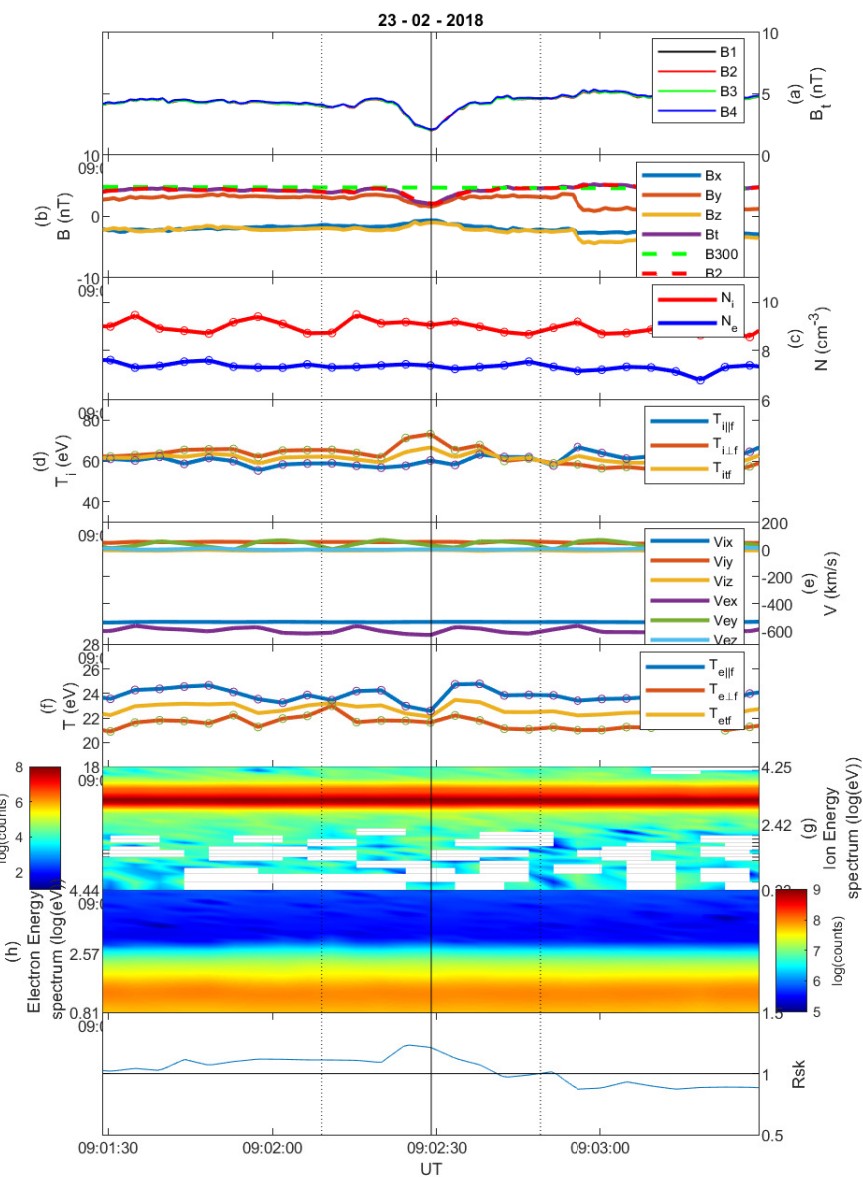

**Figure 8.** Linear magnetic hole on 23 February 2018, category "hot". Format as in Fig. 1.

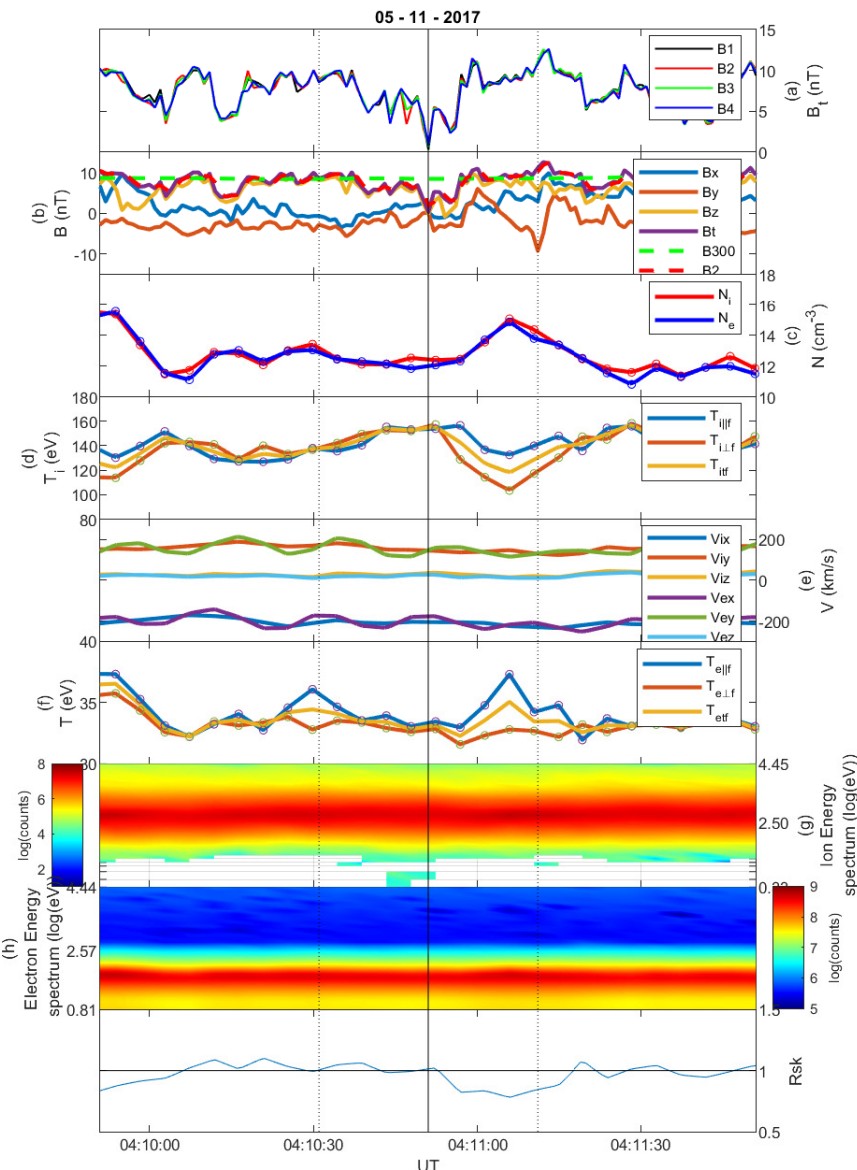

**Figure 9.** Linear magnetic hole structure on 5 November 2017, category "sign-change". Format as in Fig. 1.

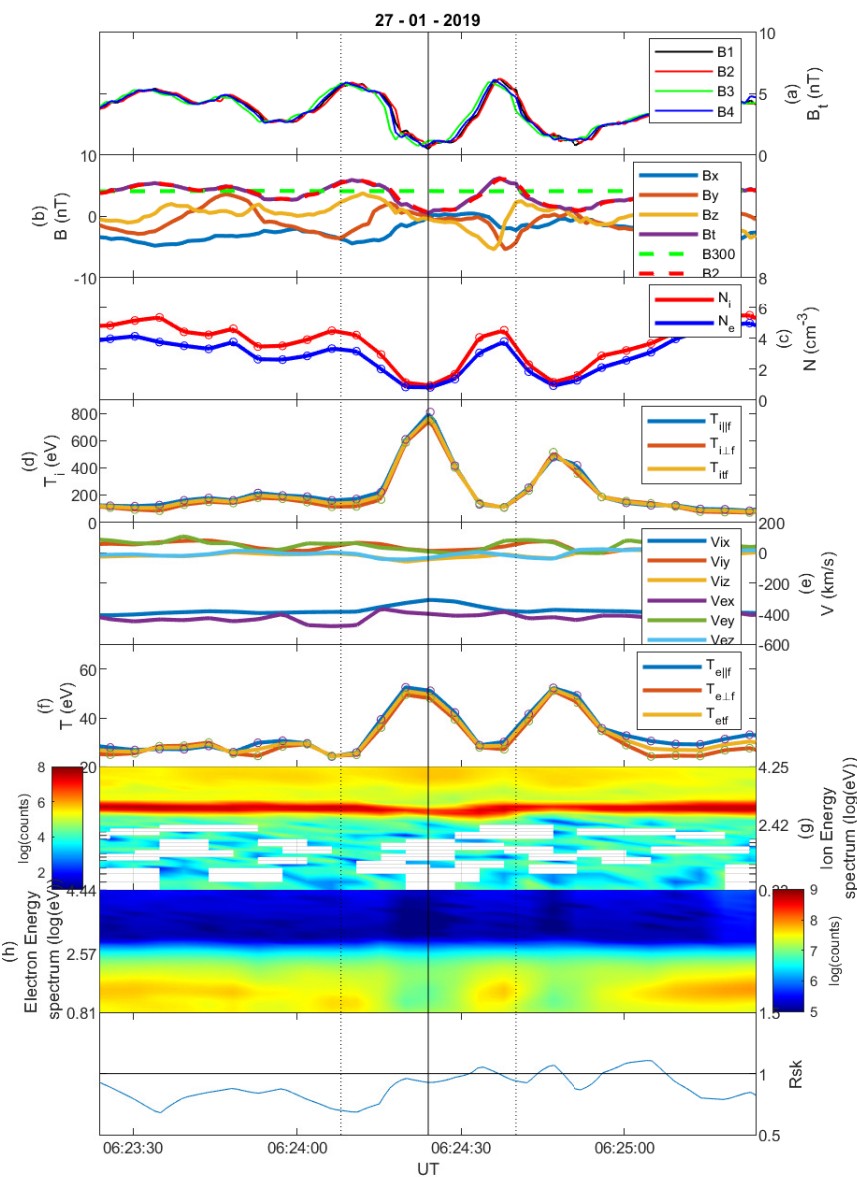

**Figure 10.** Linear magnetic hole on 27 January 2019, category "sign-change". Format as in Fig. 1.

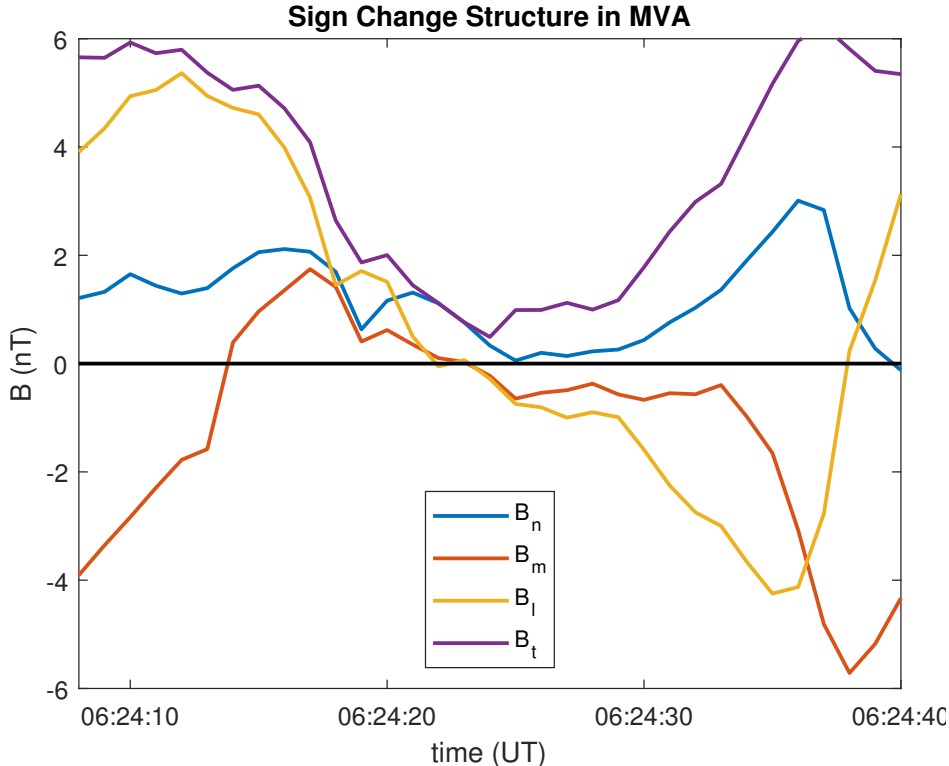

**Figure 11.** Sign-change hole of Fig. 10 transformed into an MVA coordinate system.

## 4 Pressure Balance

As discussed in the introduction, these LMH structures are assumed to be in pressure balance with their surroundings (Burlaga and Lemaire, 1978; Burlaga et al., 1990; Winterhalter et al., 1995). The decrease in magnetic pressure needs to be balanced by an increase in plasma pressure, which means a density or temperature increase.

In order to study the pressure balance of these structures the magnetic pressure and the plasma pressure are calculated: inside at the centre of the structure, and outside the average over two intervals of 30 s before and and after the structure is calculated,

similar to the determination of the magnetic field outside the LMHs. In Fig. 12 (a) the relation between the total pressure outside and inside is presented in green circles. The black line shows the identity, on which the points should lie for perfect pressure balance and perfect instrumentation. The red line shows a linear fit to the green points, with the regression coefficient and the slope listed in the figure. It is clear that there is a spread in the points around the identity. Fig. 12 (b,c,d) show the relations for the different categories, cold, hot and sign-change, respectively.

The cold LMHs show that, apart from 5 structures, there is almost perfect pressure balance, demonstrated by the slope of 1.01 of the linear fit to the points. The five exceptions, three above and two below the identity, are influenced by hot ions in the

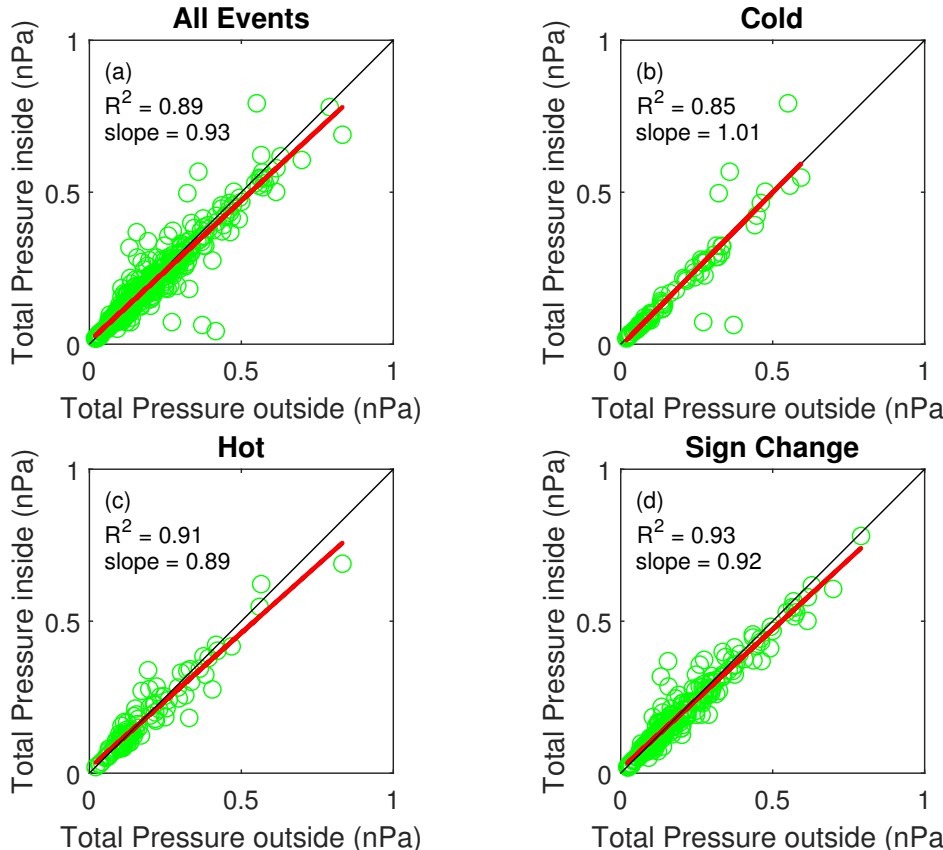

**Figure 12.** Test of the presssure balance over the structures. The total pressure is calculated at the centre and outside of the structure. (a) Relation for all structures, the black line is the identity, red line is a fit to the points. Then follow pressure balances for (b) the cold category, (c) the hot category, and (d) the sign-change category. The regression coefficients and the slopes are listed in the panels.

foreshock region. For the hot and sign-change LMHs the spread around the identity is larger and the slope of the fit deviates more from the identity.

## 5  Categories Revisited

Three categories of MHs are defined above, however, this alone does not clear up the differences between them, e.g. how and where each category is created. For the sign-change LMHs it was determined that these are mainly foreshock structures. Fig. 13 shows, for each category, the location of the structures and the direction of average background magnetic field projected onto the $XY$-plane.

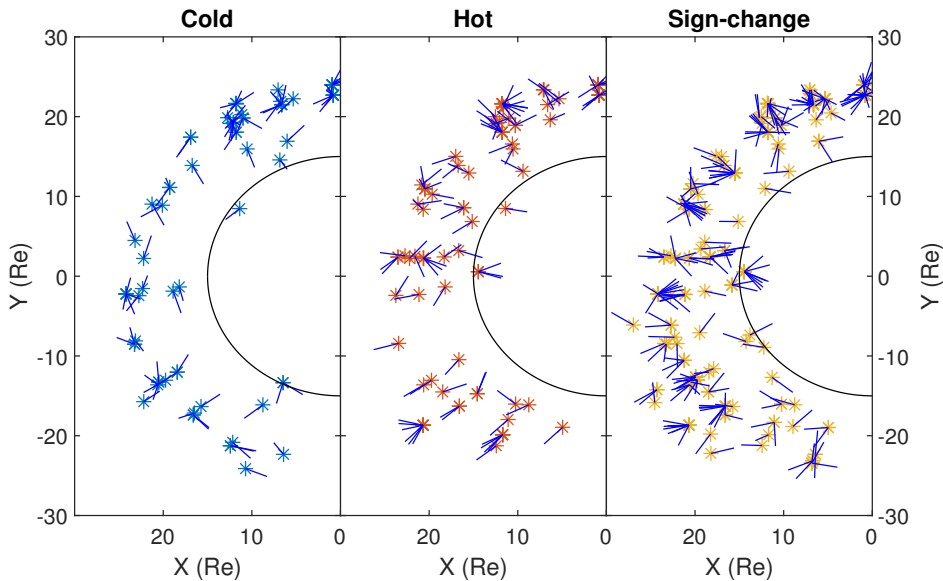

**Figure 13.** Location of the events in the GSE $XY$-plane with the average magnetic field direction for the three categories. The black circle represents the $15 R_{\mathrm{E}}$ boundary outside of which the LMHs are searched for.

In order to find out if the events are connected to the bow shock the angle between the magnetic field direction and the radial direction to the spacecraft is calculated. Fig. 14 shows the percentage of events in bins of $10°$ between the radial direction and the magnetic field direction. (Note that the angles are folded around $90°$ as the magnetic field can point in two directions.)

Fig. 14 shows that for the "cold" LMHs the distribution increases strongly for larger angles. With $\sim 20\%$ events with an angle $\theta_{\mathrm{BR}} \le 50°$ it can be concluded that these structures have basically no connection to the bow shock, i.e. are not influenced by the foreshock region.

For the "hot" and "sign-change" LMHs there is basically the same distribution of angles, with $\sim 50\%$ events with an angle $\theta_{\mathrm{BR}} \le 50°$, indicating that a much greater part of these structures can have a connection to the bow shock and can be influenced by foreshock processes.

As it is often suspected that MHs are the final stage of MMs, the instability criterion $R_{\mathrm{SK}}$, Eq. (1), is determined inside and outside of the structures. Outside, the mean value over 30 s before and after the structure is determined (similar to the average magnetic field) and inside the value at the centre of the LMH is used. The criterion $R_{\mathrm{SK}}$ in the middle of the structure vs. outside of it is plotted in Fig. 15. The "cold" (43) and "hot" (155) LMHs for the "sign change" category also have been determined and plotted as red and blue dots respectively, both of them in the sign change panel, and per category in the cold and hot panels. This shows that most of the "cold - sign change" LMHs occur for $R_{\mathrm{SK,out}} < 1$, the "hot - sign change" LMHs form a cloud similar to the green circles.

For the plasma to be MM-unstable it is required that $R_{\mathrm{SK}} > 1$. It is clear from Fig. 15 that there is a large group of structures that lie beyond the line $R_{\mathrm{SK}} = 1$ on both axis. The percentages of stable LMHs are given in Table 2.

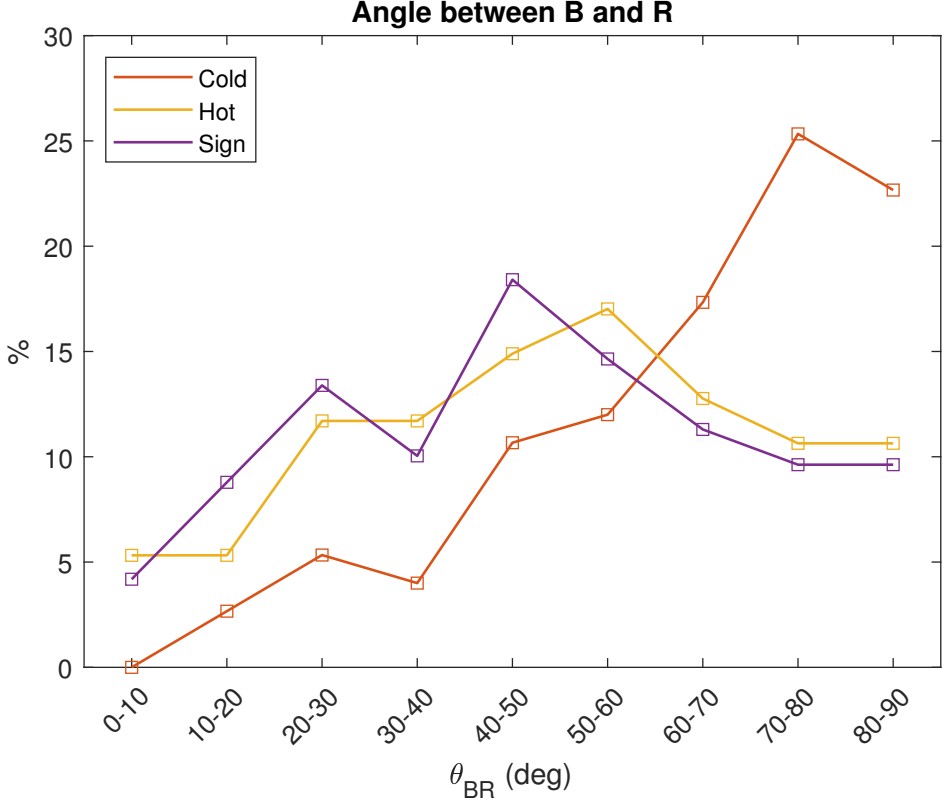

**Figure 14.** The angle between the radial direction and magnetic field direction binned per category in $10°$-wide bins as a percentage of the total number of structures per category.

A note should be made here about the instability criterion $R_{\mathrm{SK}} > 1$, which depends on the ion density and temperature, see Eqs.(1) and (2). Because of the observed discrepancies between the ion data from FPI and Wind (Bandyopadhyay et al., 2020) or OMNI (Roberts et al., 2021) one should cautious interpreting the results shown in Fig. 15 and in Table 2.

The percentages shown in Table 2 for LMHs embedded in MM-stable plasma and MM stability inside the LMHs are quite low, something that was also found by Madanian et al. (2020) using MAVEN data in the solar wind upstream of Mars.

## 6 Discussion and Conclusions

The characteristics of the ubiquitous magnetic hole structures, that are indicative of temperature asymmetries in space plasmas, were studied just outside the Earth's bow shock ($R \geq 15R_{\mathrm{E}}$) with the MMS1 spacecraft. Naturally, due to the dynamics of the bow shock (see e.g., Meziane et al., 2014) $15R_{\mathrm{E}}$ may not suffice in some cases, which will then be magnetosheath structures.

---

[1] AKA Explorer 43

[2] After exclusion of the "sign change" category this number reduces to 0.9

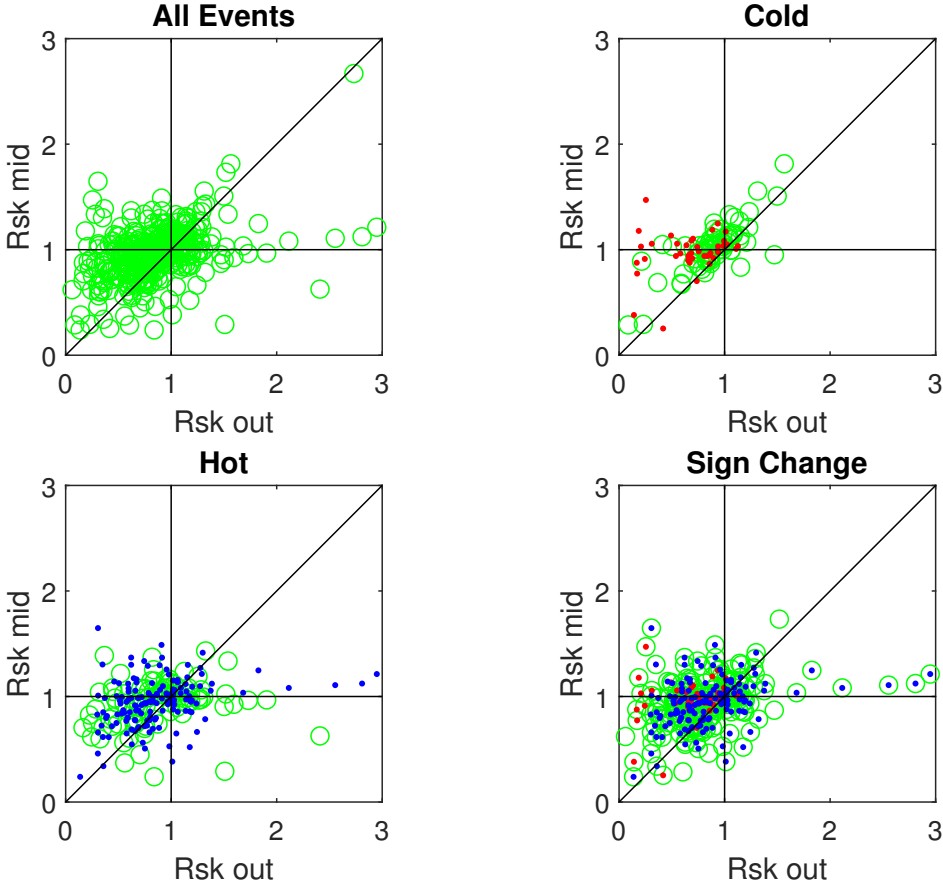

**Figure 15.** The MM instability criterion of Eq. (1), $R_{\mathrm{SK}}$, inside the LMH vs. outside. The full list of structures and the three categories are shown. The black horizontal and vertical lines show where $R_{\mathrm{SK}} = 1$, and the diagonal black line is the identity. The red (43) and blue (155) dots are the "cold" and "hot" LMHs in the "sign change" category.

**Table 2.** Top: Numbers and percentages of MM-stable conditions outside and inside of the LMHs per category. Bottom: Numbers and percentages of MM-stable LMHs in stable outside plasma and of MM-stable LMHs in unstable outside plasma.

| category | $R_{\mathrm{SK,out}} < 1$ | % | $R_{\mathrm{SK,mid}} < 1$ | % |
|---|---|---|---|---|
| Cold | 49 | 65% | 29 | 39% |
| Hot | 66 | 70% | 56 | 60% |
| Sign Change | 179 | 75% | 147 | 62% |
| category | $R_{\mathrm{SK,out}} < 1$ & $R_{\mathrm{SK,mid}} < 1$ | % | $R_{\mathrm{SK,out}} > 1$ & $R_{\mathrm{SK,mid}} < 1$ | % |
| Cold | 25 | 33% | 4 | 5% |
| Hot | 39 | 41% | 17 | 18% |
| Sign Change | 126 | 53% | 18 | 7% |

A test has been performed on the closest bin, $15 - 16R_{\mathrm{E}}$, to see how many of the 22 events were in the magnetosheath, which resulted in only three events, all on the same day.

Because of the large size of the MHs in the solar wind and the small interspacecraft distance only the data from one MMS spacecraft are analyzed. The top panel in Fig. 1 and consecutive data figures show how little difference exists between the magnetic field data. This also means that the usual 4-spacecraft analysis methods, such as timing and curlometer techniques (Schwartz, 1998) cannot be applied here. Only a significant difference between the spacecraft can be observed in the case of sub-ion magnetic holes (Wang et al., 2020c, b). A set of 406 structures were found with $\Delta B/B > 0.5$, and a maximum rotation of the magnetic field of $\Delta\theta \leq 10°$, over a time span of 8 months in 2017 and 2018. This leads to an initial occurrence rate of 2.3 per day, which is slighly higher than earlier presented rates.

After inspection of the combined magnetic field and plasma data, the LHM structures were split up into three categories:

- "Cold" LMH: A decrease of the magnetic field strength, combined with an increase in density which ensures pressure balance over the structure (75 structures);

- "Hot" LHM; A decrease of the magnetic field strength, combined with an increase in temperature with little density variation (94 structures);

- "Sign-change" LMH: A decrease of the magnetic field strength, combined with the change of sign of at least one of the magnetic field components (237 structures, of which 43 are "cold" and 155 are "hot").

If only the first two categories are counted as LMHs, as the "sign change" might be foreshock structures, then the occurrence rate listed in Table 1 should be reduced by a factor 0.4 leading to a rate of 0.9 per day. This would be lower than what has been observed near Earth in previous studies as listed Table 1. Using Cluster data, Xiao et al. (2014) showed in their table 1 occurrence rates of 0.8 and 1.1 per day for 2003 and 2004, which were also years during the declining phase of the solar cycle, in the same way as 2017 and 2018 for the present study with MMS. Also, the conditions for LMHs that Xiao et al. (2014) used ($B_{\mathrm{min}}/B \leq 0.75$ and less than $15°$ rotation of the field over the LMD) are less stringent than in this paper.

Fig. 5 shows that the density variation seems to be limited to $-1 < \Delta N/N = (N_{\mathrm{in}} - N_{\mathrm{out}})/N_{\mathrm{out}} < 1$. This means that the limit on the density inside, for the structures in this study, is $N_{\mathrm{in}} < 2N_{\mathrm{out}}$. Similarly the temperature is limited $\Delta T/T = (T_{\mathrm{in}} - T_{\mathrm{out}})/T_{\mathrm{out}} > -1$, but there should be no real upper limit. For the cases in this study $T_{\mathrm{in}} < 6T_{\mathrm{out}}$. These specific limiting values are, most likely, the result of the pressure balance of these structures, as shown in Fig. 12.

One main result is that the structures are all basically in pressure balance with their surroundings, as is clearly shown in Fig. 12. This was also found by Madanian et al. (2020) for some events in Mars's extended exosphere, where the ion temperature increased, and thus would fall into the "hot" category of this paper.

Another main result concerns the MM instability criterion, i.e. $R_{\mathrm{SK}} > 1$. In this study only $\sim 54\%$ of all structures, though varying by category, have $R_{\mathrm{SK,mid}} < 1$, and thus are MM-stable. Winterhalter et al. (1995) found that the MHs mainly occurred in a (marginally) stable plasma environment. In this study $\sim 70\%$ of the structures are embedded in an MM-stable plasma environment, $R_{\mathrm{SK,out}} < 1$, and $\sim 10\%$ of the structures in an MM-stable environment are MM-unstable inside.

Would one not expect stability inside the structure if the MHs are the final stage of MMs, for which the instability criterion should have been relaxed through the creation of the MMs? One reason for a temperature asymmetry in the MHs is the presence of an ion/electron vortex in the MH, the presence of which was shown by Wang et al. (2020b, a). Why less than half of the structures are still (or again) MM-unstable needs to be further investigated, e.g. by numerical simulations, to find the temporal evolution of MHs. A dedicated ISSI team on the topic "Towards a Unifying Model for Magnetic Depressions in Space Plasmas" will study this topic further.

*Data availability.* The data were obtained from the MMS Science Data Center (https://lasp.colorado.edu/mms/sdc/public/).

*Author contributions.* Volwerk, Goetz and Plaschke were the instigators of this project. Mautner did the preliminary data search as an intern. Simon Wedlund, Karlsson, Schmid and Rojas-Castillo helped with programming and interpreting the various results from the data analysis. Roberts and Varsani were taken into the project to help with the interpretation of the FPI data.

*Competing interests.* The authors declare that they have no conflict of interest.

*Financial support.* Charlotte Goetz is supported by an ESA Research Fellowship. Cyril Simon Wedlund is supported by the Austrian Science Fund (FWF) under project N32035-N36. Daniel Schmid was supported by Austrian Research Promotion Agency (FFG) ASAP MERMAG-4 under contract 865967.

*Acknowledgements.*

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
