# Peer review of "Statistical study of linear magnetic hole structures near Earth"

_Annales Geophysicae, 2020_

## Referee Comment (RC1) · Anonymous Referee #1 · 3 Aug 2020

The manuscript discusses the interesting topic of linear magnetic holes or magnetic depressions upstream of Earth. These structures have attracted much attention recently as their generation mechanism is still a point of debate. Authors use MMS data, which are new, and provide a new point of view on categorizing these structures based on their internal plasma properties. However, I have a few concerns regarding data usage that I think should be considered and clarified.

Major comments:

Authors indicated that they use MMS data when the orbit apogee was in the upstream solar wind. However, the FPI instrument was often shut down beyond 20 R_E in the solar wind even though the FGM instrument might have been in operation. So, I am a little doubtful about the statistics beyond 20 R_E. Have the authors checked whether

for each event in their study there are simultaneous FPI measurements? Automated computer programs sometimes select the closest available data point when there is a gap in the data.

In addition, the FPI instrument is not a solar wind monitor, and it is not optimized to measure cold plasmas such as the solar wind. Therefore, it tends to underestimate the solar wind density, while overestimating the temperature. Although this issue can perhaps be justified by the fact that it exists for all events; nonetheless, I think a statement must be added to the text where you introduce the data ($\sim$Line 48), reminding readers of this issue, and that the conclusions are drawn under such condition.

Similarly, differences in density and velocity of electrons versus ions, and between burst versus fast mode data, as shown in the figures, are instrument effects. I suggest using ion moments to present the solar wind plasma density and velocity.

After reading through the manuscript, it appears that the third category, "sign change", of magnetic holes are mostly foreshock events (e.g. HFAs, foreshock cavities). Are you suggesting a new term for these structures? Foreshock anomalies are not related to mirror mode waves and classifying them as magnetic holes seems like mixing two different types of plasma phenomena.

Here are some other minor comments:

Line 46: Do you mean 2017/18?

Line 64: "This resulted in 426 LMH", Please specify if these events are down selected from a larger dataset, or for how many of these events FPI data are available.

Line 66-70: Are you using fast mode or burst mode data? I think it is the former. If that is the case, then it should be mentioned here explicitly. I also suggest removing the burst data that are overplotted on some of the figures. If data from both modes are used, then some explanation on how they are used together is expected here.

Line 74 and Fig. 2: You display different types of LMHs in Fig. 2, but you have not

introduced them yet. Maybe consider moving this figure to the end of the section.

Line 87-90: Have you considered including additional conditions similar to Criterion 4 in Madanian et al. (2020) in your search algorithm to exclude these structures?

Fig. 9 and the caption: These measurements are made in the magnetosheath, behind the bow shock, not in the foreshock.

Fig. 10: As you described in the Introduction section, you are not interested in fore-shock events. Figures 10 and 11 obviously show foreshock events. So what is the rationale for these figures? They don't seem to add much context to your objectives.

Line 201: This time period is different than the one mentioned in the Introduction section? Did you include 2018/19 data?

Line 205: The event breakdown in different categories does not quite add up to the total of 426 events.

––––––––––––––––––––––––––––––––––

---

## Referee Comment (RC2) · Anonymous Referee #2 · 8 Oct 2020

**Review:**

This paper aims to discuss a statistical analysis of so-called 'linear magnetic hole' structures in the solar wind upstream from the Earth's bow shock by using MMS data.

They use some phenomenological criteria to characterize these structures in comparison with previous works on the same topic.

These structures are worth being studied since they are often related to the mirror instability in the plasma and they can also have some geoeffectiveness for the dynamics of the downstream magnetosheath and magnetosphere.

The results shown in the present paper are interesting and certainly deserve to to be shown to the interested community. However there is a strong need for improvement before that as is detailed below in the following report.

In particular, though the authors seem to have a strong experience on analyzing magnetic field data only, they are also using plasma particle data in the present study with apparently a little expertise on them and their analysis is thus not always well-conducted which has led to some caveat and some misinterpretation sometimes. This should be corrected before any publication. The present analysis leads the authors to define three different types of structures. But, to my opinion, these may also have different nature for some of them compared to what has been already published in the literature in particular inside the foreshock. This may lead to some controversy sometimes but controversy is an important part of science so I let the author propose their interpretation to the community if they can provide good arguments.

Therefore, the paper deserves to be published in Annales Geophysicae but only after some modification from the following comments, questions and suggestions which are related to order in the manuscript and not to importance:

1. Page 2, Introduction, line 24: About Equation (1), note that it applies only if Bi-Maxwellian velocity distributions can be assumed. Generalized instability criterion for any velocity distribution has been discussed by Hellinger (2007):

Hellinger, P. (2007), Comment on the linear mirror instability near the threshold, Phys. Plasmas, 14, 082105, doi:10.1063/1.2768318.

It could be mentioned that the classical bi-Maxwellian approximation is usually relevant for the plasma conditions here.

2. Page 3, lines 52-54: The bow shock location can vary a lot depending on external solar wind conditions and mainly on its dynamic pressure. Though distances larger than 15 Re should

generally be enough to avoid to be inside the shock region (or the magnetosheath), have the authors ensure that it is always the case in their data set especially for time intervals where the dynamic pressure is very low? The variation of the bow shock location has been studied for instance recently by Meziane et al. (2014).

Meziane, K., T. Y. Alrefay, and A. M. Hamza (2014), On the shape and motion of the Earth's bow shock, Planet. Space Sci., 93–94, 1–9.

Since mirror modes are very often observed in the magnetosheath, using such approach could help to suppress such intervals (if any).

3. Page 4, line 76: The width w is an apparent temporal width in the observations (in the time series). This is not an intrinsic physical scale and depends obviously on the spacecraft velocity and the dynamics of the structure in case it is also varying with time. So I would rephrase it such as 'The apparent temporal width w of the LHM structures in the time series is …'

4. Page 4, line 81-85: This comparison hold for one single spacecraft assuming that the structure does not evolve in size with time and does not propagate itself with respect to the ambient solar wind plasma (stationary structure). MMS is a four-spacecraft mission so basically it should help to disentangle between spatial and temporal variation in principle. I do not understand why this is not addressed here! At least a comparison between the four time series is mandatory (at least for B magnitude). Is the temporal width nearly constant between the four MMS satellites? It should be possible to quickly check whether the structures propagates with a finite velocity in the solar wind frame with the classical multi-spacecraft techniques (e.g. Schwartz, 1998) but considering the spacecraft locations and the time of the center of each structure for instance. Maybe the small spatial separation is a strong limitation and make this determination unconclusive. But this should be at least discussed and quantitatively shown for any clear case such as the ones shown on Figure 6 for instance.

Schwartz, S. (1998), ISSI Scientific reports series 1, 249, http://www.issibern.ch/pdf-Files/analysis_methods_1_1a.pdf

5. Page 4, line 84: This is the thermal Larmor radius. It should be written.

6. Page 4, line 89: Hot flow anomalies are not the only kind of foreshock transients which can be observed. What about e.g. the foreshock cavities (e.g. Sibeck et al., 2002), density holes (Parks et al., 2006), foreshocks cavitons (Blanco-Cano et al., 2009; 2011; Kajdič et al., 2013)? Some of the characteristics of these transients are common with the phenomenological criteria used here to characterize the magnetic holes. It is mandatory to clearly explain the differences between them. This could be made in the introduction for instance. But it would be very helpful for the

reader to explain how the structures (so-called linear magnetic holes) which are the topic of the present study clearly differ from all the already described foreshock transients (apart the fact that they can be observed outside the foreshock of course).

Blanco-Cano, X., N. Omidi, and C. T. Russell (2009), Global hybrid simulations: Foreshock waves and cavitons under radial interplanetary magnetic field geometry, J. Geophys. Res., 114, A01216, doi:10.1029/2008JA013406

Blanco-Cano, X., P. Kajdič, N. Omidi, and C. T. Russell (2011), Foreshock cavitons for different interplanetary magnetic field geometries: Simulations and observations, J. Geophys. Res., 116, A09101, doi:10.1029/2010JA016413

P. Kajdič, X. Blanco-Cano, N. Omidi, K. Meziane, C. T. Russell, J.-A. Sauvaud, I. Dandouras, B. Lavraud, Statistical study of foreshock cavitons, Annales Geophysicae, 10.5194/angeo-31-2163-2013, 31, 12, (2163-2178), (2013).

Parks, G. K., Lee, E., Mozer, F., et al.: Larmor radius size density holes discovered in the solar wind upstream of Earth's bow shock, Phys. Plasmas, 13, 050701, 2006.

Sibeck, D. G., Phan, T.-D., Lin, R., Lepping, R. P., and Szabo, A.; Wind observations of foreshock cavities: A case study, J. Geophys. Res., 10, 1271, doi:10.1029/2001JA007539, 2002.

Sibeck, D. G., Kudela, K., Mukai, T., Nemecek, Z., and Safrankova, J.: Radial dependence of foreshock cavities: A case study, Ann. Geophys., 22, 4143–4151, 2004, http://www.ann-geophys.net/22/4143/2004/

7. Page 5, Figure 1: There are many things to say about this Figure and this case:

a) first, there is an inversion in panel (a) in the legend between By and Bt (for Btotal I guess, which is the magnitude of B since it is not explicit neither in the text nor in the caption, and different from the t,n,m components on line 154 for instance). I guess the purple curve which is always negative here is By and the red curve is B total. It seems to be the case for other Figures like 6 and 7.

b) there is an obvious issue on panel (b) since the ion and electron densities shown are different. This is systematic and even worse on Figure 6 (with nearly 50% difference!). Since quasi-neutrality must be ensured in the solar wind plasma at the considered scales, there is a need for some explanation here! I notice that no co-author of the present paper is a member of the particle experiment teams. At least the authors should have ask them why there is this obvious

discrepancy! So I guess the data are like that on the database as is often the case. It is also mentioned that people who want to use the data must be aware of the possible caveats and be ready to ask the expertise of the instrument teams. There has been no cross-calibration made apparently. It is well-known that it is always difficult to provide a correct absolute density from an electron spectrometer (which is not designed for that!). It comes from the moment calculations. As is well-known, the most important issue is to correctly take into account the spacecraft potential, usually positive in the solar wind due to spacecraft photoelectrons and which tends to perturb the low energy measurements (where most of the density is). This usually leads to underestimate the density. It also strongly influence the temperature moment while the velocity moment is always much more difficult to compute than for the ions due to the large electron thermal velocity.

The description of the ion and electron data set is quite short in the present paper. I would strongly suggest to add a better description in part 2.

c) the other issue for panel (b) is that the behavior of the ion and electron densities (whatever the time resolution) is different for the time where the magnetic field magnitude depression is observed in the center of the Figure. Electron density seems correlated while ion density seems mostly anti-correlated! Moreover the ion density is strongly varying around the magnetic hole with some irregular fluctuations (ULF waves?) on a scale similar to the magnetic field depression (where it is the least variable). The authors should comment on that.

d) how are the different ion temperature components on panel (c) computed? Do they come from a diagonalization of the pressure (or temperature) tensor or are they computed in a frame whith one axis along the ambient magnetic field (average on a larger scale) using the magnetic field data so that the parallel and perpendicular temperatures are the real true ones? This is very important since the definition of the parallel and perpendicular temperature then strongly depends on the eigenvalues after the diagonalization. If one eigenvalue is largely separated from the two others, that's fine and it is assumed that the corresponding eigenvector is the magnetic field direction giving the parallel temperature and the mean of the two other eigenvalues provide a good estimate of the perpendicular temperature. But when the three eigenvalues are quite close, the software always gives these two temperatures but they may have no physical meaning in terms of real temperature anisotropy.

Also, some clear ULF fluctuation is observed mostly on the blue curve (parallel temperature for ion fast mode) for all the time interval shown (and maybe on a larger one?) which make any conclusion about its variation where the magnetic depression is observed quite delicate.

e) There is a big issue with the electron velocity components shown on panel (d). There is strong regular modulation on the main component (Vex) and also on Vey. This also observed

systematically for nearly all the other cases shown in the following figures (6, 7, 8, 9 and 10). This too regular fluctuation (of the order of 20 seconds) seems to be an artefact of the moment calculation for a reason I cannot infer (it does not seem to be related to a spacecraft spin period effect?). Otherwise there would be a permanent fluctuating current in the plasma (quite odd) which is not physically possible. Interestingly, the peak of the Vex curve seem to be correlated with the ion density maxima except around the magnetic depression. But the authors should ask the experimenters about this if they want to show these data. Is the electron velocity data really useful in the present study?

It would be interesting to show the electron spectrogram time series (like panel (f) for the ions) to see whether such modulation is visible or not. It may be systematically shown for all cases.

f) the authors should explain why there is so much difference between the ion and electron fast and burst mode temperature sets on panels (c) and (e). This could be made earlier (part 2) if there is an explanation about this. Which data set is the most accurate? Are the burst (high time resolution) data necessarily better? This should have been discussed in the experimenters' first papers which should be at least referenced (maybe by reproducing what they wrote about this point). For the temperature calculation, the most important apart the field of view coverage of the instrument is the angular resolution used. When this angular resolution is not the best possible, it tends to overestimate the temperature from the 3-D velocity distribution. It is often better for onboard calculations for a 3-D distribution which is rarely transmitted in the telemetry at a high cadence. Moreover, I guess that the ion temperature used does not discriminate the core solar wind proton population from the hotter alpha (He++) population. This automatically leads to overestimate the temperature. If there is no possibility (from what the experimenters provide) to separate the two populations, then the thermal pressure supposed to be for the main population (protons) will be usually overestimated (depending on the density ratio n_alph/np) since the ion temperature Ti > Tp (sometimes by a large amount) and the pressure will be computed as np*Ti while it should be np*Tp + n_alph*T_alph with T_alph > Tp (by a factor between 2 and 8 typically) but n_alph<<np. Only when n_He++/np is very small there is no issue (then Ti ~Tp). But for a case like the one shown on Figure 6 where a clear He++ peak is visible, that could lead to some error in the pressure determination for calculating the beta for instance.

Also, I do not see any specific signature of the magnetic hole in these panels. Is it really useful to show these data for this case?

I get the feeling the authors systematically display all the data they have available without any specific purpose and without a very good knowledge of the particle data they use, letting the reader trying to understand the observations. I understand that the temperature data are important later to compute the thermal pressures (with the limitations I already mention on the moment

accuracy). But does this mean that showing their time series is mandatory here if they do not display anything clear?

Another general comment: the labels are very small and not easy to read. Also it should be always mentioned for which MMS spacecraft the data come from (both in the text and in the caption).

8. Page 7, line 119: I do not agree that the two temperature are 'basically the same' 'Outside'. This is true before the magnetic hole but not after.

9. Page 7, line 112: How can the authors infer that the mirror instability criterion is fulfilled without simply checking it? At least the increase of Tperp is consistent with it but not sufficient.

10. Page 7, line 114: The two cases shown on Figure 6 are much more clear than the previous one on Fig. 1. To me there are the typical linear magnetic holes observed inside a time interval with very steady magnetic field! Again, I would suggest to suppress the electron bulk velocity data. Here the electron temperature data show very interesting features.

Again the observations are shown for only one spacecraft. I get the feeling that the clear observations here make possible the identification of the entry and exit times from each structure or the central time (by using 1-s running average for instance to smooth the data around the minimum). Then I would like to see a Figure with one panel for each component Bx,y,z and |B| for the 4 MMS spacecraft with different colors to check if they differ or not. If they do, it should be possible for instance to check whether these structure are consistently non propagating in the solar wind plasma and if they are stationary (in width in particular). Again maybe the error bars can be too large with the small inter-distances but it needs to be checked at least!

11. Page 8, line 125: How can the authors state that? There is a clear need for an elaborated reasoning before concluding that the structure are merging. For instance, is there any support from a simulation result to propose this? It would also be interesting to check whether the time separation between the two holes in the times series differ for the other MMS satellites?

12. Page 9, line 131: I suggest to write '… the structures are ' likely ' MM-unstable.' or to give the result of the instability criterion (equation 1) to be able to state that (same remark as my point (9) above).

13. Page 9, lines 132-133: There is also a correlation of the electron temperature with the magnetic field magnitude depression while the ion temperature is clearly anti-correlated for this case. Again this is a clear nice case in term of magnetic field observation. But for the ion density

(apart for the quasi-neutrality issue already mentioned), there is a clear ambient low frequency fluctuation with a quasi-period of about 20 s again and an amplitude of about 0.5 cm-3 which seems damped for the time interval of the magnetic hole. What is the origin of this fluctuation? It seems for this case when looking to the ion density maxima to be clearly correlated with the modulation of the electron velocity moment in Vex and Vey which really again clearly looks like an experimental artefact (when looking to Vey on Figure 6, mostly Vex and Vey on Figure 7, 8, 9).

14. Page 9, line 139: Is this case really in the solar wind? The ion spectrogram display a very hot distribution which is clearly not the pristine solar wind. Same for the bulk ion velocity which seems very low (around 200 km/s). As I already mentioned, I strongly recommend the author to check whether this is not a case in the magnetosheath by looking also to the electron spectrogram for instance (electron variation is strong at the shock) and/or to check with a bow shock location model taking into account the (true) upstream solar wind pressure (from another spacecraft like ACE for instance). Models like Cairns et al. (1995) could be useful for that:

Cairns, I. H., D. H. Fairfield, R. R. Anderson, V. E. H. Carlton, K. I. Paularena, and A. J. Lazarus (1995), Unusual locations of Earth's bow shock on September 24–25, 1987: Mach number effects, J. Geophys. Res., 100, 47–62.

15. Page 13, line 142: There seems to be another small case before 04:1150 UT. Is there a reason (from the selection criteria) not to select it? Also, it would be very nice to add vertical dashed lines surrounding the tile interval where the magnetic hole structure is supposed to be identified (also on other Figures like that).

16. Page 13, line 145: The ion spectrogram does not reveal a superimposition of clear solar wind population plus a very energetic (backstreaming ions) component as it should be in the ion foreshock. I totally disagree with the authors there. See my point (14).

17. Page 13, line 146-149: Here I agree this very nice observation is inside the ion foreshock and moreover inside the ULF foreshock waves boundary since clear typical nonlinear '30-s ULF waves' are seen both on the magnetic-field and the ion density. The ion spectrogram clearly reveals the waves on the solar wind (red peak) with a clearly separated ion foreshock population. The ion velocity (which seems to be correctly computed contrary to the electron one) also displays the effect of these waves. So my question: how this structure shown here differs from the so-called 'cavitons'?

18. Page 13, line 159: This sentence seems strange here since the pressure balance has not been yet proven! It should be rephrased. This paragraph is devoted to the study of the (possible) pressure balance. The conclusion about any study should be given after the analysis not before.

19. Page 13, line 168: Have the authors check the reason why these 5 cases are outliers? Since it is a small number considering these individual cases should be very quick.

20. Page 16, line 176-178: There is something I do not understand here (and I guess some kind of reasoning error about the foreshock). Basically this angle on Figure 14 deals with the radial component of the magnetic field. When considering Fig. 13 giving the locations of the observations which are all on the dayside, it seems obvious that a nearly radial field will always intercept the bow shock surface. But for a larger angle, this will strongly depends on WHERE is the spacecraft in space since the bow shock does not obviously have a spherical shape. Moreover, its shape and location depends on the solar wind conditions which are not always the same. In general, one needs to check connection to the bow shock surface at the observation point using a bow shock model again. A large ThetaBr angle (not radial field) does not always prevent the extrapolated field line to intercept the shock (conic) model especially when the spacecraft is close to the nose of the bow shock or the X_GSE axis. Even a nearly orthoradial field line can be connected to the shock (and thus the spacecraft being in the foreshock by definition). So I would be very careful with the sentence written line 179-181 which for me is not conclusive. For the methodology to check whether the spacecraft is located in the foreshock or not, I suggest the authors to look studies like e.g.:

Meziane and d'Uston (1998). A statistical study of the upstream intermediate ion boundary in the Earth's foreshock. Ann. Geophysicae 16, 125-133.

Mazelle, C., et al. (2003). Production of Gyrating Ions from Nonlinear Wave-Particle Interaction Upstream from the Earth's Bow Shock: A Case Study from Cluster-CIS, Planetary and Space 505 Science, 51, 785–795, https://doi.org/10.1016/S0032-0633(03)00107-7

Meziane, K., et al. (2004), Bow shock specular reflected ions in the presence of low frequency electromagnetic waves: a case study, Annales Geophys. 22: 1–11, SRef-ID: 1432-0576/ag/2004-22-1

Eastwood, et al. (2006). The foreshock, Space Sci. Rev., 11, 41–94.

21. Page 21, line 224: Could the author provide any reference to explain why this is expected? There is no mention of any theoretical work on the nonlinear evolution of the mirror mode instability in the present paper.

Minor points (just to help the authors):

- Page 4, line 87: ' the Earth and' its 'bow shock'

- Page, 6 line 97: Upper delta T at the end of the line
- Page 13, line 162: 'center'

---

## Author Comment (AC1) · 11 Nov 2020

*Authors indicated that they use MMS data when the orbit apogee was in the upstream solar wind. However, the FPI instrument was often shut down beyond 20 R_E in the solar wind even though the FGM instrument might have been in operation. So, I am a little doubtful about the statistics beyond 20 R_E. Have the authors checked whether for each event in their study there are simultaneous FPI measurements? Automated computer programs sometimes select the closest available data point when there is a gap in the data.*

During the automatic search for the magnetic holes, the availability of FPI data was not one of the criteria. However, for all events found in the automated search FPI data was available, so there was no down-selection on plasma data.

*In addition, the FPI instrument is not a solar wind monitor, and it is not optimized to measure cold plasmas such as the solar wind. Therefore, it tends to underestimate the solar wind density, while overestimating the temperature. Although this issue can perhaps be justified by the fact that it exists for all events; nonetheless, I think a statement must be added to the text where you introduce the data (~Line 48), reminding readers of this issue, and that the conclusions are drawn under such conditions.*

Indeed, the referee makes a good point here, and this should have been mentioned in the paper. The FPI instrument has a "solar wind" mode, but that does not work perfectly for the ions. We have added two plasma specialists (Roberts and Varsani) to the author list and we describe the problem with the FPI instrument. From a recent paper by Owen Roberts it was shown, through comparison of FPI and OMNI data, that the electron density is well determined, whilst in general, the ion density is under estimated (although there are exceptions, as is clear from the events shown in the current paper and from the figures in the Roberts et al. paper).

*Similarly, differences in density and velocity of electrons versus ions, and between burst versus fast mode data, as shown in the figures, are instrument effects. I suggest using ion moments to present the solar wind plasma density and velocity.*

We have decided to not include the burst mode data, as these are not available for all events. The differences in density and velocities are instrumental effects, as mentioned above. However, from the statistical study between FPI and OMNI it follows that the electrons should be favoured.

*After reading through the manuscript, it appears that the third category, "sign change", of magnetic holes are mostly foreshock events (e.g. HFAs, foreshock cavities). Are you suggesting a new term for these structures? Foreshock anomalies are not related to mirror mode waves and classifying them as magnetic holes seems like mixing two different types of plasma phenomena.*

Indeed, as stated in the paper, we consider the "sign change" events as possible foreshock structures and thus in the end the possibility is brought up to not consider them as magnetic holes, reducing the occurrence rate of MHs. And no we are not suggesting a new term for these structures, that would be counter-productive for the space physics community in which a multitude of different names for the same structure leads more to confusion than clarification.

*Line 46: Do you mean 2017/18?*

Indeed, this should have been 2017/18

*Line 64: "This resulted in 426 LMH", Please specify if these events are down selected from a larger dataset, or for how many of these events FPI data are available.*

As mentioned above, for all 406 (not 426, that is a typo that was continued throughout the paper) FPI data was available. There was no down-selection based on the presence of FPI data.

*Line 66-70: Are you using fast mode or burst mode data? I think it is the former. If that is the case, then it should be mentioned here explicitly. I also suggest removing the burst data that are overplotted on some of the figures. If data from both modes are used, then some explanation on how they are used together is expected here.*

We have decided to only use the fast mode data as burst mode is not available for all 406 events. The partially available burst mode data do not add anything specific to the analysis.

*Line 74 and Fig. 2: You display different types of LMHs in Fig. 2, but you have not introduced them yet. Maybe consider moving this figure to the end of the section.*

Here, in the text we present the occurrence rate of the full selection of the 406 MHs and plot them as a function of distance from Earth. In the text is it said that the other colours are described further down, which we think we can expect the reader of the paper to understand.

*Line 87-90: Have you considered including additional conditions similar to Criterion 4 in Madanian et al. (2020) in your search algorithm to exclude these structures?*

No, we have not considered that, because in the end we characterize the "sign-change" category as foreshock structures, which is also clear from the examples that are shown.

*Fig. 9 and the caption: These measurements are made in the magnetosheath, behind the bow shock, not in the foreshock.*

Indeed, this event is rather strange, as from the ion spectrogram it looks like magnetosheath. The location of this event, however, is at (5, 22, 3) Re, a rather far distance ~23 Re to be in the magnetosheath. The solar wind is unremarkable at ~380 km/s (with little Vy and Vz) and ~3/cc density (OMNI) and the subsolar bow shock distance is ~14.5 Re. On the MMS website the locator software also puts the spacecraft in the magnetosheath. This has been corrected in the paper.

*Fig. 10: As you described in the Introduction section, you are not interested in foreshock events. Figures 10 and 11 obviously show foreshock events. So what is the rationale for these figures? They don't seem to add much context to your objectives.*

These two figures are there to show examples of third category, to show the reader what they look like.

*Line 201: This time period is different than the one mentioned in the Introduction section? Did you include 2018/19 data?*

This is a typo, and has been corrected in the text. The time period is like described at the beginning of the paper.

*Line 205: The event breakdown in different categories does not quite add up to the total of 426 events.*

As mentioned above, this typo has continued itself throughout the paper. There are 406 events.

---

## Author Comment (AC2) · 11 Nov 2020

*Page 2, Introduction, line 24: About Equation (1), note that it applies only if Bi-Maxwellian velocity distributions can be assumed. Generalized instability criterion for any velocity distribution has been discussed by Hellinger (2007). It could be mentioned that the classical bi-Maxwellian approximation is usually relevant for the plasma conditions here.*

The referee is correct that the bi-Maxwellian approach is usually appropriate for situations as discussed in the current paper, and thus the widely used instability criterion is given in the text. We have added a small comment that a more general approach can be found in Hellinger (2007).

*Page 3, lines 52-54: The bow shock location can vary a lot depending on external solar wind conditions and mainly on its dynamic pressure. Though distances larger than 15 Re should generally be enough to avoid to be inside the shock region (or the magnetosheath), have the authors ensure that it is always the case in their data set especially for time intervals where the dynamic pressure is very low? The variation of the bow shock location has been studied for instance recently by Meziane et al. (2014). Since mirror modes are very often observed in the magnetosheath, using such approach could help to suppress such intervals (if any).*

This would be a possibility, but that would mean to start over completely. We do, however, check a small sample of events for their location to get an estimate of the percentage of magnetosheath events.

*Page 4, line 76: The width w is an apparent temporal width in the observations (in the time series). This is not an intrinsic physical scale and depends obviously on the spacecraft velocity and the dynamics of the structure in case it is also varying with time. So I would rephrase it such as 'The apparent temporal width w of the LHM structures in the time series is …'*

The text has been revised accordingly.

*Page 4, line 81-85: This comparison hold for one single spacecraft assuming that the structure does not evolve in size with time and does not propagate itself with respect to the ambient solar wind plasma (stationary structure). MMS is a four-spacecraft mission so basically it should help to disentangle between spatial and temporal variation in principle.*

Yes, one would think that the 4-spacecraft mission would help entangle spatial and temporal variations through the methods developed for the Cluster mission. Unfortunately the inter-spacecraft separation of MMS is so small that the structures we are looking at in this paper are too large, and all four spacecraft basically see the same thing. The Bt for all four spacecraft is added to the data figures, and that shows that e.g. timing analysis is not possible. However, it can be done for sub-ion scale magnetic holes, where there is a clear distinct feature at each spacecraft (Wang et al., 2020).
This was already mentioned in the Discussion and Conclusions section of the paper, line 198. However, we have extended this comment a bit further.

*Page 4, line 84: This is the thermal Larmor radius. It should be written.*

The text has been revised accordingly and also the velocity is now called the thermal velocity.

*Page 4, line 89: Hot flow anomalies are not the only kind of foreshock transients which can be observed. What about e.g. the foreshock cavities (e.g. Sibeck et al., 2002), density holes (Parks et al., 2006), foreshocks cavitons (Blanco-Cano et al., 2009; 2011; Kajdič et al., 2013)? Some of the characteristics of these transients are common with the phenomenological criteria used here to characterize the magnetic holes. It is mandatory to clearly explain the differences between them.*

Indeed, there are many more foreshock structures than just HFAs. We have added the various structures that the referee mentions in this comment. However, we think that it is well beyond the scope of this paper to discuss these structures in detail.

*Page 5, Figure 1: There are many things to say about this Figure and this case:*
*a) first, there is an inversion in panel (a) in the legend between By and Bt (for Btotal I guess, which is the magnitude of B since it is not explicit neither in the text nor in the caption, and different from the t,n,m components on line 154 for instance). I guess the purple curve which is always negative here is By and the red curve is B total. It seems to be the case for other Figures like 6 and 7.*

Unfortunately, the colours of the lines got disturbed through the inclusion of burst mode data (blue on top of red turns into a purple), which have now been removed from the paper. The use of Bt (as in Btotal) has been used as later a minimum variance coordinate system is used in the paper with l, m and n components. It would thus be confusing to use Bm for the magnitude of the magnetic field and Bt is used.

*b) there is an obvious issue on panel (b) since the ion and electron densities shown are different. This is systematic and even worse on Figure 6 (with nearly 50% difference!). Since quasineutrality must be ensured in the solar wind plasma at the considered scales, there is a need for some explanation here!*
*…*
*The description of the ion and electron data set is quite short in the present paper. I would strongly suggest to add a better description in part 2.*

This is an omission in the paper, as there is no discussion about the fact that the MMS FPI instrument was not developed for solar wind conditions, even though the instrument has a solar wind mode. The differences that are seen in e.g. the densities of the protons and electrons are purely instrumental. We have added two plasma specialists to our team: Owen Roberts and Ali Varsani, of whom the first has statistically studied the behaviour of FPI in the solar wind by comparing the data sets with OMNI data (something similar was done for the ARTEMIS mission by Artemyev et al, 2018). We have added a discussion about this in the paper. The electron densities are in good agreement with the OMNI data, the ion densities are, statistically, under-estimated (where of course there are exceptions, such that the ion density is over estimated).

*c) the other issue for panel (b) is that the behavior of the ion and electron densities (whatever the time resolution) is different for the time where the magnetic field magnitude depression is observed in the center of the Figure. Electron density seems correlated while ion density seems mostly anti-correlated! Moreover the ion density is strongly varying around the magnetic hole with some irregular fluctuations (ULF waves?) on a scale similar to the magnetic field depression (where it is the least variable). The authors should comment on that.*

The ion density is varying with a ~20-s period, similar to the variations seen in the electron velocity in panel (e). This is the spacecraft spin tone, which has not been removed correctly from the data.
We have added a comment on the spin tone in the paper.

*d) how are the different ion temperature components on panel (c) computed? Do they come from a diagonalization of the pressure (or temperature) tensor or are they computed in a frame whith one axis along the ambient magnetic field (average on a larger scale) using the magnetic field data so that the parallel and perpendicular temperatures are the real true ones?*
*…*
*Also, some clear ULF fluctuation is observed mostly on the blue curve (parallel temperature for ion fast mode) for all the time interval shown (and maybe on a larger one?) which make any conclusion about its variation where the magnetic depression is observed quite delicate.*

This is beyond the scope of this paper and for readers interested in this specific knowledge the references to the instrument papers are in the text.
The purported ULF waves are the spacecraft spin tone.

*e) There is a big issue with the electron velocity components shown on panel (d). There is strong regular modulation on the main component (Vex) and also on Vey. This also observed systematically for nearly all the other cases shown in the following figures (6, 7, 8, 9 and 10). This too regular fluctuation (of the order of 20 seconds) seems to be an artefact of the moment calculation for a reason I cannot infer (it does not seem to be related to a spacecraft spin period effect?).*
*…*
*It would be interesting to show the electron spectrogram time series (like panel (f) for the ions) to see whether such modulation is visible or not. It may be systematically shown for all cases.*

The 20-s signal that is seen is the spacecraft spin tone.
The electron time-energy spectrograms have been added to all data figures.

*f) the authors should explain why there is so much difference between the ion and electron fast and burst mode temperature sets on panels (c) and (e).*

As there is no burst data for most of the events in this paper, these have been taken out of the paper. This choice is recalled in Section 2.

*Also, we do not see any specific signature of the magnetic hole in these panels. Is it really useful to show these data for this case? We get the feeling the authors systematically display all the data they have available without any specific purpose*

We are surprised by this comment, as all the data that we show are actually used to calculate the physical size of the MHs (velocity) in order to scale them to the local Larmor radius (B, N, T).
In the first event, figure 1, there is a signature in the ion temperature, which is shorter than the 20-second spin tone and aligns well with the magnetic hole. This might have been hidden through the addition of the burst mode data, which have now been removed altogether to avoid confusion.

*Another general comment: the labels are very small and not easy to read. Also it should be always mentioned for which MMS spacecraft the data come from (both in the text and in the caption).*

We will take care that the labels are larger in the final version of the paper. Also, a mention is now made in the paper that only the data from MMS1 are used.

*Page 7, line 109: I do not agree that the two temperature are 'basically the same' 'Outside'. This is true before the magnetic hole but not after.*

The text in the paper was not correct, and this paragraph has been rewritten, also in view of the fact that the burst mode data have been taken out of the paper.

*Page 7, line 112: How can the authors infer that the mirror instability criterion is fulfilled without simply checking it? At least the increase of Tperp is consistent with it but not sufficient.*

The referee is correct and an extra panel with the instability criterion has been added to the figures, using the condition Rsk from Eq. (1).

*Page 7, line 114: The two cases*
*…*
*Again the observations are shown for only one spacecraft.*

The reason for this has been explained above.

*Page 8, line 125: How can the authors state that? There is a clear need for an elaborated reasoning before concluding that the structure are merging. For instance, is there any support from a simulation result to propose this? It would also be interesting to check whether the time separation between the two holes in the times series differ for the other MMS satellites?*

This comment follows from MHs developing out of MMs. If the Bohm-like diffusion, proposed by Hasegawa & Tsurutani, works on two neighbouring MMs, then through growth of these to MMs they

might merge and create this "double dipped" structure. Therefore is it is stated in the text that COULD be an indication. More text is added in the discussion section in order to describe what we were thinking here.

*Page 9, line 131: I suggest to write '… the structures are ' likely ' MM-unstable.' or to give the result of the instability criterion (equation 1) to be able to state that (same remark as my point (9) above).*

We have added the instability criterion to the figures and added a comment in the text.

*Page 9, lines 132-133*

Fluctuations have been explained above

*Page 9, line 139: Is this case really in the solar wind? The ion spectrogram display a very hot distribution which is clearly not the pristine solar wind. Same for the bulk ion velocity which seems very low (around 200 km/s). As I already mentioned, I strongly recommend the author to check whether this is not a case in the magnetosheath by looking also to the electron spectrogram for instance (electron variation is strong at the shock) and/or to check with a bow shock location model taking into account the (true) upstream solar wind pressure (from another spacecraft like ACE for instance).*

This is a strange case. The bulk velocity is about 300 km/s (both Vx and Vy are ~200 km/s). The MMS spacecraft are located at (4, 21, 0.2) Re, the solar wind conditions are nominal from Wind B~3.5 nT with positive Bz, Vx ~380 km/s, ni ~ 4/cc, dynamic pressure ~1.1 nPa and an Alfven Mach number of ~10, with a bow shock sub solar point at ~14.5 Re.
It is not impossible that this is in the magnetosheath, the spacecraft seems to be at least near the nominal bow shock as shown in Cairns et al. [1995]. But the MMS website shows that the spacecraft is (deep) in the magnetosheath.
Indeed, checking at the MMS website, the spacecraft locator shows the spacecraft in the magnetosheath.

https://lasp.colorado.edu/mms/sdc/public/plots/#/historical-orbit?year=2020&month=11&day=09&time=10:00:00&plot_type=XY
Taking the set of the nearest events, i.e. R < 16 Re, only 3 (all on the same day) out of 22 are in the magnetosheath.

*Page 13, line 142: There seems to be another small case before 04:1150 UT. Is there a reason (from the selection criteria) not to select it? Also, it would be very nice to add vertical dashed lines surrounding the tile interval where the magnetic hole structure is supposed to be identified (also on other Figures like that).*

The figures have been adapted, to have the vertical lines in all panels, that was a mistake in the plotting routine. And yes, because of the selection method some hole may not be selected. This has been explained in Volwerk et al., 2020, and this results in an underestimation of about 10%. This has been added to the text describing the second event.

*Page 13, line 145: The ion spectrogram does not reveal a superimposition of clear solar wind population plus a very energetic (backstreaming ions) component as it should be in the ion foreshock. I totally disagree with the authors there. See my point (14).*

Yes, this might indeed be the magnetosheath, see above.

*Page 13, line 146-149: Here I agree this very nice observation is inside the ion foreshock and moreover inside the ULF foreshock waves boundary since clear typical nonlinear '30-s ULF waves' are seen both on the magnetic-field and the ion density. The ion spectrogram clearly reveals the waves on the solar wind (red peak) with a clearly separated ion foreshock population. The ion velocity (which seems to be correctly computed contrary to the electron one) also displays the effect of these waves. So my question: how this structure shown here differs from the so-called 'cavitons'?*

In Fig 11 we have transformed the data into a MVA coordinate system, which shows that this structure is possibly a flux rope or a hot flow anomaly. The cavitons as in Kajdic et al. (2013) have durations greater than approximately 1 minute (see their table 1), which is much longer than the structure in Figs. 10 and 11. We see no evidence for cavitons to have a flux rope structure in Kajdic et al. (2011, 2013).

*Page 13, line 159: This sentence seems strange here since the pressure balance has not been yet proven!*

True, we should have stated that the structures "are assumed to be in pressure balance", now corrected.

*Page 13, line 168: Have the authors check the reason why these 5 cases are outliers?*

We have checked the 5 "outliers" (and updated the figure). The text has been adapted and the reason is that these are influenced by the foreshock.

*Page 16, line 176-178: There is something I do not understand here (and I guess some kind of reasoning error about the foreshock). Basically this angle on Figure 14 deals with the radial component of the magnetic field. When considering Fig. 13 giving the locations of the observations which are all on the dayside, it seems obvious that a nearly radial field will always intercept the bow shock surface.*

Yes, the referee is correct, this deals with the "radial component" of the magnetic field, insofar that the direction of the magnetic field around the MHs is determined, and the angle of B with the radial direction from the Earth's centre to the spacecraft is determined. Figure 14 shows that for the "cold" category the events are mainly observed at a large angle, which means that they are unconnected to the bow shock/foreshock region. In the other categories, it looks slightly different, with smaller populations at large angles.

*Page 21, line 224: Could the author provide any reference to explain why this is expected? There is no mention of any theoretical work on the nonlinear evolution of the mirror mode instability in the present paper.*

It is our understanding that when an instability is triggered in a plasma, this leads to a relaxation of the instability criterion outside of the formed structure. However, there are no papers (as far as we know) that show that inside the MH the instability criterion should not be fulfilled. However, there are papers that show electron and ion vortices in MHs, which would indicate a temperature asymmetry. This has been added to the paper, including this topic being the centre of an ISSI team, which will first meet next year.

Page 4, line 87: ' the Earth and' its 'bow shock'

corrected

Page, 6 line 97: Upper delta T at the end of the line

Corrected

Page 13, line 162: 'center'

We prefer British spelling for a European journal.

---

## Referee Report (RR1)

**Review:**

The authors have mostly answered correctly to the numerous questions I raised.

In particular, they address much better the issues in the particle data used from MMS (with for instance the spin tone of 3 RPM not removed which was obvious in the Figures). I fully understand that they cannot correct these parameters by themselves and have to use what is provided with the identified caveats. However it was mandatory to well explain this limitations to any reader who should have been quite 'shocked' by the strong discrepancy between electron and ion density. A better job could have been done (but this clearly beyond the scope of the present study) if correct cross-calibrated particle data could have been systematically determined (maybe with the help of the instrument team) by using the numerous missions in the solar wind around the Earth like Wind as is mentioned for instance or ACE.

After saying that, I still have some concern about the quality of the determination of the mirror mode instability criterion (Rsk) which is used in the paper. This parameter obviously depends both on the ion density and temperature which are said to be both affected by the mentioned instrumental effects. Maybe the underestimation of the density is compensated by the overestimation of the temperature in the computation of the beta parameter and the temperature anisotropy is correctly determined, but it is mandatory that the authors add a small sentence about this and keep some caution about the determination of the Rsk parameter. It should be mentioned when dealing with Table 2 and Figure 15. This will not in general change their conclusions and prevent them from publishing their analysis.

So the new version of the manuscript is now suitable for publication in Annales Geophysicae provided that my last comments and suggestions are taken into account.

Typo: line 129: 'only changes a little'.

---

## Author Response (AR2)

[revised manuscript text omitted]

We would like to thank the referees for their time reviewing this paper for the second time.

We thank referee #1 for accepting the paper as is.

We agree with referee #2 that there can be done quite a lot still on calibrating the FPI data in the solar wind. But indeed, this falls out of the scope of this paper and is mainly a task for the MMS FPI team, although Roberts et al. (2021) have at least started to look into this topic.

A note has been added in the text, when discussing Fig. 15 and Table 2 that because of discrepancies that have been observed in the FPI data that caution should be used interpreting the results here at lines 217 − 219.

As for the typo on line 129, I prefer "has changed little" over "has changed a little", I will leave that to the editorial team of AG.